# Evolution of herbivore-induced early defense signaling was shaped by genome-wide duplications in *Nicotiana*

Wenwu Zhou[1], Thomas Brockmöller[1], Zhihao Ling[1], Ashton Omdahl[1,2], Ian T Baldwin[1], Shuqing Xu[1]*

[1]Department of Molecular Ecology, Max Planck Institute for Chemical Ecology, Jena, Germany; [2]Brigham Young University, Provo, United States

**Abstract** Herbivore-induced defenses are widespread, rapidly evolving and relevant for plant fitness. Such induced defenses are often mediated by early defense signaling (EDS) rapidly activated by the perception of herbivore associated elicitors (HAE) that includes transient accumulations of jasmonic acid (JA). Analyzing 60 HAE-induced leaf transcriptomes from closely-related *Nicotiana* species revealed a key gene co-expression network (M4 module) which is co-activated with the HAE-induced JA accumulations but is elicited independently of JA, as revealed in plants silenced in JA signaling. Functional annotations of the M4 module were consistent with roles in EDS and a newly identified hub gene of the M4 module (NaLRRK1) mediates a negative feedback loop with JA signaling. Phylogenomic analysis revealed preferential gene retention after genome-wide duplications shaped the evolution of HAE-induced EDS in *Nicotiana*. These results highlight the importance of genome-wide duplications in the evolution of adaptive traits in plants.

## Introduction

Induced defense is widespread in plants and can improve the fitness of plants under herbivore attack (*Baldwin, 1998*; *Kessler et al., 2004*). Many plants recognize and distinguish the damage caused by feeding insects from mechanical damage by perceiving herbivore-associated elicitors (HAE) to induce rapid early defense signaling (EDS) that includes the accumulation of jasmonic acid (JA) and its derivatives, phytohormones that play a central role in the activation of induced defenses (*Erb et al., 2012*; *Howe and Jander, 2008*; *Wu and Baldwin, 2010*). Increases or decreases in leaf JA concentrations can directly activate or impair induced anti-herbivore defenses, respectively (*Farmer and Ryan, 1992*; *Kessler et al., 2004*; *Wu and Baldwin, 2010*), highlighting the importance of JA accumulation for induced defenses. However, increased JA levels can also reduce plant fitness due to the physiological and ecological costs of defense elicitation when defenses are not needed (*Baldwin and Hamilton, 2000*; *Glawe et al., 2003*; *Heil and Baldwin, 2002*; *van Dam and Baldwin, 1998*). For example, in *Nicotiana attenuata*, an increase in endogenous JA levels by supplying methyl-jasmonic acid (MeJA) reduced plant fitness by 26% when plants were protected from herbivore attack (*Baldwin, 1998*). Thus induced JA accumulations can result in net fitness gains or losses depending on the cost/benefit ratio of induced defenses, which varies among attacking herbivore species and environmental conditions. Therefore, a robust and complex signaling network that regulates and fine-tunes induced JA biosynthesis, metabolism and JA-dependent induced downstream defenses is essential for plants to realize their fitness optima.

Using reverse genetics, such as RNA-inference (RNAi) and virus induced gene silencing (VIGS), several genes that are rapidly induced by HAE were found to regulate JA biosynthesis and metabolism in plants, particularly in the wild tobacco *Nicotiana attenuata* which has been established as an

*For correspondence: sxu@ice.mpg.de

**eLife digest** A variety of different insects feed on plants and these insects often produce molecules known as elicitors that the plants can recognize. This triggers a sophisticated suite of defenses in the plant that can either deter feeding by the insects, or help the plants endure the attack. The elicitors stimulate the rapid accumulation of a plant hormone called jasmonic acid, which in turn activates the defense responses. However, high levels of jasmonic acid can also reduce the ability of the plants to survive and reproduce by activating plant defenses when they are not needed. Therefore, plants need to regulate the signaling networks that control defense so that jasmonic acid only accumulates when the benefit of fighting the insect outweighs the cost of producing the defenses.

The costs and benefits of defense responses vary among different insects and environmental conditions, which has made it difficult to study how plants regulate defense signaling networks. To address this question, Zhou et al. investigated the activities of genes in six species of tobacco plant after they have been exposed to different insect elicitors.

The experiments identified a network of genes that is activated in response to elicitors and acts largely independent of jasmonic acid signaling. A newly identified gene in this network called NaLRRK1 and jasmonic acid suppress each other, suggesting that NaLRRK1 helps to regulate jasmonic acid levels. Further analysis shows that a process called genome duplication, in which all the genes in an organism are copied, has shaped the evolution of early defense signaling in *Nicotiana*. Many of the duplicated genes have adopted new roles and been retained in the plants. This highlights the importance of genome duplications in helping plants to adapt to their environment.

The next challenge following on from this work would be to identify what specific roles these genes play in the plants, and how they affect the ability of plants to survive insect attacks in their native habitats.

ecological model system for plant-herbivore interactions (*Wu and Baldwin, 2010*). The HAE-regulated signaling network includes: protein kinases, such as the wounding induced protein kinase (*NaWIPK*) (*Wu and Baldwin, 2010*; *Wu et al., 2007*) and calcium-dependent protein kinases (*NaCDPK4/5*) (*Yang et al., 2012*), which positively and negatively regulate HAE-induced JA accumulations, respectively; transcription factors, such as *NaWRKY3/6*, which positively regulate HAE-induced JA accumulations (*Skibbe et al., 2008*); and ethylene (ET) biosynthesis and perception genes (*NaETR1, NaACO* and *NaACS*) (*von Dahl et al., 2007*), which crosstalk with JA-regulated downstream defense responses (*Onkokesung et al., 2010*; *Voelckel et al., 2001*), such as nicotine biosynthesis (*Kahl et al., 2000*; *Shoji et al., 2000*). While these studies have provided mechanistic insights into induced defenses, they also revealed the complexity of the HAE-induced EDS network. A systematic investigation of its complete genetic architecture is essential to understanding the molecular mechanisms and evolution of HAE-induced EDS.

Gene duplications play a key role in network evolution (*Pastor-Satorras et al., 2003*; *Teichmann and Babu, 2004*). Duplicated genes can either be retained in the same network to increase network complexity and robustness or evolve to function in new networks through subfunctionalization and/or neofunctionalization processes (*De Smet and Van de Peer, 2012*; *Duarte et al., 2006*) that can be detected from changes in the spatiotemporal expression or protein interaction patterns of the duplicated genes. Although both gene expression and protein-protein interaction divergences between duplicated genes increase over time (*Arabidopsis Interactome Mapping, 2011*), several factors, such as the type of duplication and the functionality of the genes, affect the rate and extent of those divergences (*Hanada et al., 2008*; *Rizzon et al., 2006*). For instance, expression divergences between duplicated genes involved in stress responses tend to be greater than those of duplicated genes involved in developmental processes (*Ha et al., 2007*). While studies based on the analysis of gene ontologies and genome-wide duplications suggest that linage-specific duplication (LD) followed by expression divergence are important for the evolution of stress responses in plants (*Hanada et al., 2008*; *Rizzon et al., 2006*), whole genome duplications (WGD)

events, which are prominent in the plant kingdom, provide a major source of duplicated genes and contribute significantly to the evolution of cellular networks, such as gene regulatory (*Blanc and Wolfe, 2004*), protein-protein interaction (*Arabidopsis Interactome Mapping, 2011*) and metabolic networks (*Gachon et al., 2005*; *Hofberger et al., 2013*). Furthermore, duplicated copies from WGD events are more likely to be retained in a network than those from LD, especially for genes that are dosage-sensitive, such as transcription factors, and protein kinases (*Arabidopsis Interactome Mapping, 2011*; *Birchler and Veitia, 2007*; *Casneuf et al., 2006*; *Edger and Pires, 2009*; *Freeling, 2009*). However, the relative contribution of WGD and LD to the evolution of HAE-induced EDS networks and the patterns of expression divergence between duplicated genes in these networks have not been studied.

Understanding the molecular mechanisms and evolution of HAE-induced EDS requires the identification of the genome-wide HAE-induced EDS networks. Because functionally related genes tend to be transcriptionally coordinated (*Persson et al., 2005*; *Stuart et al., 2003*), co-expression network analysis has been widely used to infer the function of genes and uncover biological pathways (*Klie et al., 2014*; *Usadel et al., 2009*; *Yonekura-Sakakibara et al., 2008*). Distinct from 'classical' gene expression analysis using genome-wide expression profiling of control and treated samples to identify 'up' or 'down' regulated genes, co-expression network analysis uses expression measurements from a large number of samples that vary in their genotype, treatment, tissue or sampling time to enhance the statistical power of the analysis (*Zhang and Horvath, 2005*). However, due to the high specificity among tissues and treatments and the speed of the HAE-elicited responses (within 30 min) (*Gulati et al., 2014, 2013*; *Kim et al., 2011*), the general co-expression network approach that uses gene expression data from different tissues or time course experiments are not particularly useful. One solution to overcome this specificity issue is to use natural variation, such as occurs amongst closely related species or different genotypes within species, to identify co-expressed gene networks (*Ardlie et al., 2015*; *Delker et al., 2010*). In the identification of HAE-induced EDS networks, the comparison of closely related species has at least two advantages over the use of different genotypes within a species: (1) their greater genetic and phenotypic diversity (*Xu et al., 2015*) which increases the power of detecting co-expressed genes; (2) their divergence times are over several millions of years which allows for the identification of evolutionarily conserved co-expression networks that are likely functionally important.

Closely-related *Nicotiana* species within the clade of *Petunioides* show highly specific HAE-induced defenses and thus provide an ideal system for identifying HAE-induced EDS networks (*Xu et al., 2015*). Our previous study revealed that a single HAE, such as the fatty acid-amino acid conjugate C18:3-Glu (FAC) – the most active elicitor found in the oral secretions of the Solanaceae specialist herbivore *Manduca sexta* ($OS_{Ms}$) larvae – elicits diverse defense responses among closely related *Nicotiana* species when added to standardized puncture wounds. In addition, a single *Nicotiana* species, such as *N. pauciflora,* showed distinct defense responses to the FAC, $OS_{Ms}$ and oral secretions from the generalist herbivore *Spodoptera littoralis* ($OS_{Sl}$) (*Xu et al., 2015*). Here we sequenced the leaf transcriptomes of six closely-related *Nicotiana* species from the *Petunioides* clade (*N. obtusifolia, N. linearis, N. acuminata, N. pauciflora, N. miersii* and *N. attenuata*) that had been induced by three different HAEs or simply wounded (induced by wounding plus water) to characterize the HAE-induced EDS networks in *Nicotiana*. We compared HAE-induced transcriptomic responses among the six species and identified a co-expression gene network that represents the HAE-induced EDS in *Nicotiana* based on three independent lines of evidence: (1) the induction of the network correlates with variation in JA accumulations both among species treated with the same HAE and within species treated with different HAEs; (2) the induction of genes in this network that are largely not dependent on induced JA accumulations; (3) the consequences of silencing a hub gene in this network for HAE-induced JA metabolism and defenses. Analysis of the evolutionary history of all genes in the EDS network revealed that preferential gene retention after the Solanaceae whole genome triplication (WGT) event shaped the evolution of HAE-induced EDS in *Nicotiana*.

## Results

### FAC-induced early leaf transcriptomic responses are highly variable among closely related *Nicotiana* species

Closely related *Nicotiana* species showed highly divergent early transcriptomic responses within 30 min of FAC elicitation (*Figure 1*), consistent with observations from metabolomic and insect performance studies (*Xu et al., 2015*). Two species, *N. obtusifolia* and *N. miersii*, which did not respond to FAC-treatments by amplifying their wound-induced accumulations of jasmonic acid (JA) within 2 hr, showed overall little induced transcriptomic responses (*Figure 1* and *Figure 1—figure supplement*

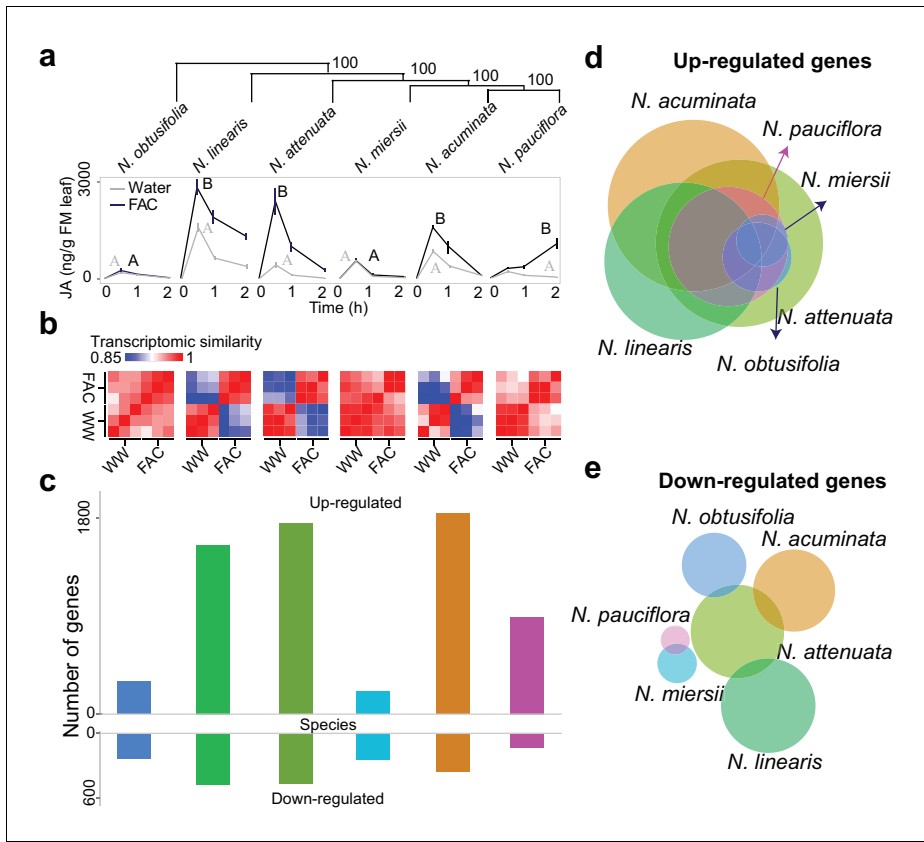

**Figure 1.** FAC elicits divergent transcriptome responses among closely related *Nicotiana* species. (a) FAC-induced JA responses among six *Nicotiana* species. Phylogenetic tree was constructed based on orthologous genes and numbers on each branch indicates bootstrap values. X-axis indicates time after elicitation and Y-axis denotes JA concentrations. FM= fresh mass. Gray and black colored lines refer to control (wounding and water) and FAC-induced samples, respectively. Different letters indicate significance between two treatments (Student's-*t* test, p<0.05). (b) transcriptomic similarity between control and FAC-induced samples (30 min after elicitation) in the six species (order is same as panel a). The color gradients indicate the Pearson correlation coefficients among samples. (c) number of differentially up- and down-regulated genes after FAC elicitation in the six species (order is same as panel a). Y-axis depicts the number of genes. Each colored bar indicates a different species. Blue: *N. obtusifolia*, light green: *N. linearis*, dark green: *N. attenuata*, light blue: *N. miersii*, orange: *N. acuminata*, pink: *N. pauciflora*. d and e, Venn diagrams of up- (d) and down- (e) regulated genes in each of the six *Nicotiana* species. Circle size indicates the relative number of up/down regulated genes in each species. Each filled circle indicates a different species, with color code as in panel c.

The following figure supplements are available for figure 1:

**Figure supplement 1.** Z-score of FAC-induced gene expression changes in six species.

**Figure supplement 2.** Validations of 12 selected FAC-induced genes in *N. attenuata*.

1). In the other four species, FAC elicitation induced both high levels of JA and significant transcriptomic changes. The variation in FAC-induced transcriptomic responses largely resulted from the upregulation of genes. Among all the species, FAC elicitation up-regulated more genes (1149.3 ± 667.8; mean ± standard deviation) than it down-regulated (353.2 ± 278.9). To validate the expression changes observed from the RNA-seq analysis, we quantified the transcript abundance of 12 *N. attenuata* genes that were found to be up-regulated by FAC using qPCR. Although two genes only showed marginally significant increases in their transcript levels (p=0.08, likely due to their large variation in expression among replicates), all 12 genes showed consistent FAC-induced up-regulation (*Figure 1—figure supplement 2*), suggesting an overall high reliability of the RNA-seq results. Interestingly, despite the large variation in FAC-induced transcriptomic responses among the species, 34 genes were induced by FAC in all six species (*Supplementary file 1A*) which likely represent the conserved FAC-induced stress response genes. Among them, eight are transcription factors from the WRKY (3), AP2/ERF (2) TT2 (1) and PLATZ (1) families, and two are E3 ubiquitin-protein ligase-like proteins (*Supplementary file 1A*). Taken together, the data suggest that while the induction of only a few genes are conserved after FAC elicitation, the overall FAC-induced early transcriptomic responses are highly variable among closely related *Nicotiana* species.

## A co-expressed gene module is induced by FAC-elicitation but not by JA

The highly divergent FAC-induced transcriptomic responses provide an excellent opportunity to identify co-expression networks. We identified FAC-induced gene co-expression networks using the weighted gene co-expression network analysis (WGCNA) method (see details in Materials and Methods). In total, five gene modules (M1-M5) were identified using control (wounding + water) and FAC-induced gene expression profiles from all six species (*Figure 2a*). Among these five modules, module M4 showed the highest correlation with HAE-induced JA accumulations, a marker of induced defense signal (*Figure 2b*). In all four species that showed FAC-induced JA accumulations (*Figure 2c*), the majority of M4 module genes were also significantly induced by FAC (p<0.05 and fold change greater than 1.5, exact negative binomial test). In contrast, in the two species, *N. obtusifolia* and *N. miersii*, which did not show FAC-induced JA accumulations (*Figure 2c*), less than 22% of the M4 module genes were induced. The intra-modular connectivity of the M4 module, a parameter that indicates the degree of co-expression among 3genes in a network, was significantly higher in FAC-induced samples than in control samples (p=0.0002, Kruskal-Wallis rank sum test), consistent with the observation that FAC elicits co-expression among genes in the M4 module (*Figure 2d*). Furthermore, the expression kinetic analysis of the M4 module genes using a previously published microarray dataset (*Kim et al., 2011*) revealed that most of the M4 module genes were largely transiently expressed (*Figure 2—figure supplement 1*) after HAE-elicitation. These data suggest that the identified M4 module is likely associated with FAC-induced EDS.

More than 53% of the M4 module genes were induced in *N. pauciflora* at 30 min after FAC-elicitation (*Figure 2b*), the time point when JA was not yet induced in this species, indicating that the induction of gene module M4 is independent of or precedes that of JA. To further test this hypothesis, using the *N. attenuata* genome-wide microarray, we measured FAC-induced gene expression changes in JA deficient *N. attenuata* plants (ir*AOC*), in which a key JA biosynthesis gene was silenced and the induced JA levels were reduced to basal levels (*Kallenbach et al., 2012*). For the comparison, we also performed genome-wide microarray analysis for the same *N. attenuata* wild type (WT) RNA samples that were used for the RNA-seq analysis. In the WT samples, fewer FAC-induced genes were detected by the microarray (771) than by the RNA-seq analysis (1752); however more than 81.2% of the FAC up-regulated genes identified using the microarray were also found from the RNA-seq analysis, indicating an overall consistency between RNA-seq and microarray data, and the expected higher power and sensitivity of RNA-seq in detecting differentially expressed genes. More than 87% of the M4 module genes that were induced by FAC in WT plants, both from the RNA-seq and microarray experiments, were also up-regulated in the JA-deficient ir*AOC* plants (*Figure 3*). Likewise, based on the microarray data of samples that were collected at 30 min after FAC-elicitation, the majority (85.1%) of up-regulated genes in WT plants were also up-regulated (FDR adjusted p<0.05, fold change >1.5) in the JA-deficient plants, suggesting that the FAC-induced early expression changes are largely not dependent on FAC-induced JA accumulations. Together, these data suggest that the genes of the M4 module are largely induced by FAC but not by JA,

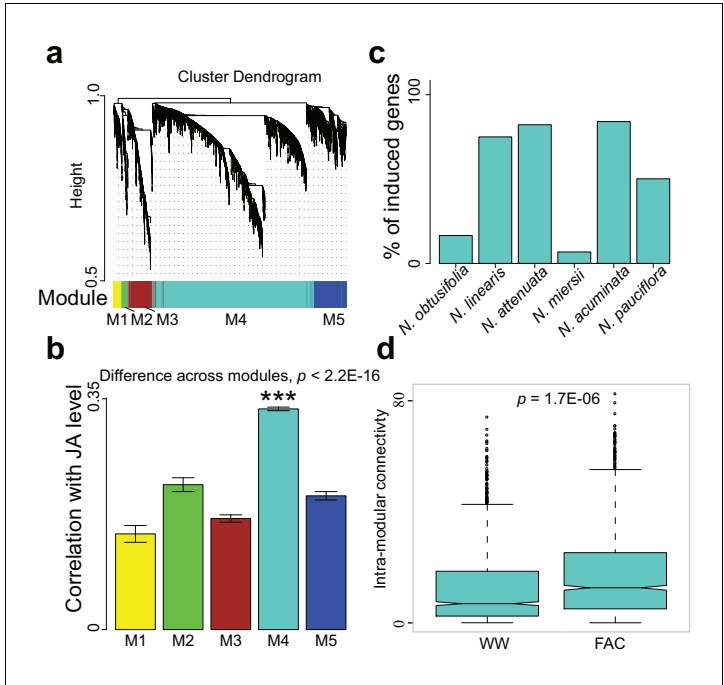

**Figure 2.** The M4 co-expression module is correlated with induced defense and is induced by FAC elicitation. (**a**) the cluster dendrogram of five modules (M1-M5). Each color indicates a different co-expression module. Y-axis indicates the height of the clustering tree. Yellow: M1, green: M2, brown: M3, turquoise: M4, blue: M5. (**b**) the average correlation coefficient between each module and the maximum induced JA level within 2 hr. Y-axis indicates the value of average correlation coefficients. Each color represents one identified module. Mean and standard error are shown for each bar. The M4 module has a significantly higher correlation coefficient than the other modules (***, p<0.001, Wilcoxon–Mann–Whitney test). (**c**) percentage of genes in module M4 induced by FAC-elicitation across six species. (**d**) the intra-modular connectivity of the M4 module in control and FAC-induced samples. WW indicates control samples (wounding and water) and FAC indicates the FAC-elicited samples.

The following figure supplement is available for figure 2:

**Figure supplement 1.** Expression kinetics of the M4 module genes based on previous microarray data.

which places their regulation down-stream of HAE perception but upstream or parallel of the activation of JA signaling.

## The induction of M4 genes is associated with the specificity of HAE-induced early defense responses within species

A previous study revealed that different HAE can induce distinct defense responses within the same species (*Xu et al., 2015*). To understand the underlying molecular mechanisms, we additionally sequenced the transcriptomes of leaves that were induced by the oral secretions (OS) of the Solanaceae specialist herbivore *M. sexta* ($OS_{Ms}$) and the generalist herbivore *S. littoralis* ($OS_{Sl}$) in four different *Nicotiana* species that showed specific responses to different HAE. This analysis revealed that the level of M4 module gene inductions correlate with the specificity of HAE-induced defense responses within a species. In *N. attenuata*, FAC, $OS_{Ms}$ and $OS_{Sl}$ induced similar levels of induced defense responses, and consistently, the majority of the M4 module genes were induced by all three elicitors (*Figure 4 a–d*). In *N. pauciflora*, while both FAC and $OS_{Ms}$ up-regulated a large fraction of the M4 module genes (*Figure 4b*) and downstream induced defense responses, $OS_{Sl}$ only up-regulated 14.8% of the M4 module genes and failed to activate the downstream defense responses (*Xu et al., 2015*). Furthermore, while in *N. obtusifolia* and *N. miersii*, both FAC and $OS_{Ms}$ only up-regulated less than 13.9% of the M4 module genes (*Figure 4b*) and did not activate the downstream defenses (*Xu et al., 2015*), $OS_{Sl}$ up-regulated 53.5% of the M4 module genes and induced

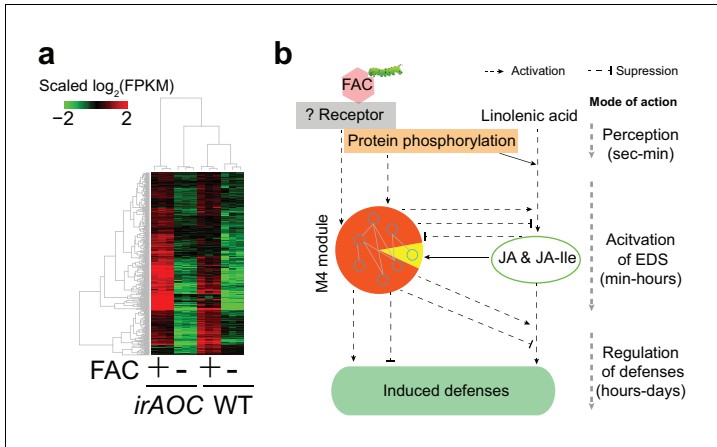

**Figure 3.** The majority of M4 module genes were induced by FAC in both WT and JA deficient plants. (**a**) a heatmap representing the expression of M4 module genes in WT and JA deficient plants (ir*AOC*). The color gradient represents the relative expression level. (**b**) a simplified induced defense signaling pathway, which indicates that accumulation of JA is not required for the induction of the majority of the M4 module (orange color in the pie chart). EDS: early defense signaling. Dashed and solid arrows indicate known and putative regulations, respectively.

downstream defense responses in these two species. These data suggest that the induction of the M4 module genes correlates with the variation of different HAE induced defense responses within species.

Although at a global level, the FAC and $OS_{Ms}$ induced similar levels of phytohormones and transcriptomic defense responses (*Figure 4a–c*), the resulting downstream defenses, such as effects on caterpillar growth rates can be different. For example, our previous study showed that larvae grew faster on leaves induced by $OS_{Ms}$ than by FAC in both *N. pauciflora* and *N. attenuata* (*Xu et al., 2015*). Because FAC is a subset of the elicitors in $OS_{Ms}$, we reasoned that $OS_{Ms}$ might contain other elicitors that suppress the downstream responses of JA accumulations (*Xu et al., 2015*). Consistent with this hypothesis, at a transcriptomic level, we found that $OS_{Ms}$ induced a smaller number of M4 genes than did FAC in both *N. attenuata* and *N. pauciflora* (*Figure 4b*). In *N. miersii*, in which both FAC and $OS_{Ms}$ did not induce defense responses. We further identified eight genes (*Supplementary file 1B*) from the M4 module that showed lower expression in response to $OS_{Ms}$ than to FAC in both *N. attenuata* and *N. pauciflora*. Among these eight genes, one gene, *NaJAR1.1* (NIATv7_g23173, *Figure 4e*), is a member of the jasmonic acid-amido synthetase (JAR1) gene family (*Figure 4f*). JAR1 catalyzes the formation of jasmonyl-isoleucine (JA-Ile), a conjugate of JA that activates downstream defense responses (*Kang et al., 2006*; *Staswick et al., 2002*). In the *N. attenuata* genome, there are three JAR1 copies that resulted from duplication events, and two of these (*NaJAR4* and *NaJAR6*) are induced by both FAC and $OS_{Ms}$ and are involved in the conjugation of JA to amino acids and anti-herbivore defense responses (*Kang et al., 2006*; *Wang et al., 2007*). Although the exact functions of *NaJAR1.1* remain unknown, it shares more than 85% of protein sequence identity to *NaJAR4* and *NaJAR6* and has the conserved amino acid conjugation domain shared by all JAR1 family members, suggesting that *NaJAR1.1* is also likely involved in the metabolism of JA. We hypothesize that an unknown component in $OS_{Ms}$, which might be used by the specialist herbivore *M. sexta* to suppress the expression of *NaJAR1.1* in order to regulate JA metabolism and thus suppress downstream defense responses in *Nicotiana*.

In summary, the induction of genes in the co-expression module M4 is associated with the specificity of HAE-induced early defense responses within species, and is consistent with the notion that induction of the M4 module is important for HAE-induced defense responses in the genus *Nicotiana*.

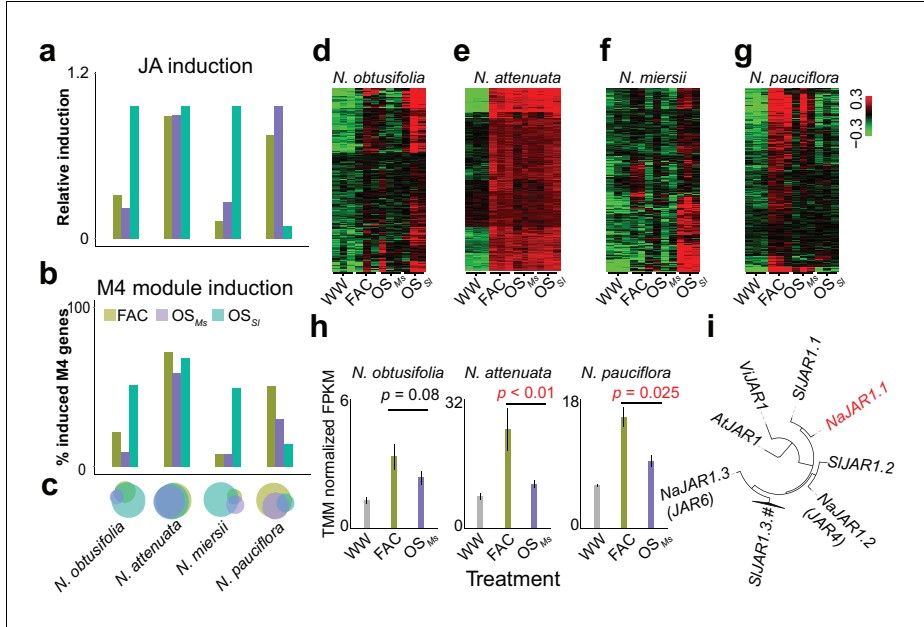

**Figure 4.** The induction of module M4 is associated with the specificity of different HAE-induced defense responses within species. a and b, the relative JA induction (a) and proportion of genes in the M4 module (b) induced by different HAEs in four *Nicotiana* species. The JA induction was scaled between 0 and 1, to indicate the lowest and highest JA level induced by three different HAEs and control (WW). Each colored bar represents elicitations from different HAEs. Dark green: FAC, purple: *M. sexta* oral secretion (OS$_{Ms}$), light blue: *S. littoralis* oral secretion (OS$_{Sl}$). (c) Venn diagrams showing the overlap among upregulated genes induced by three different HAEs within each species. Each color represents one HAE with the same color code as in panels a and b. The sizes of the circles represent the total number of genes in each group. (d–g) heatmaps showing the expression of M4 gene module members in four species as induced by the three HAEs and control. The color gradient represents the relative expression value. (h) OS$_{Ms}$ induced lower expression level of *JAR1.1* than did FAC in three *Nicotiana* species. Each bar presents the average expression (TMM normalized FPKM) of *JAR1.1* in each species. Each color indicates different treatments. Gray: control (wounding and water), dark green: FAC, purple: OS$_{Ms}$. (i) the phylogenetic tree showing the relationship among three paralogs of *JAR1* in *N. attenuata* and orthologues of *JAR1* in *Arabidopsis thaliana (At), Vitis vinifera (Vi)*, and *Solanum lycopersicum (Sl)*.

## The M4 module represents the conserved herbivore-induced EDS network among different *Nicotiana* species

The specific responses induced by different HAEs within a species also provided an opportunity to further examine the conservation of the M4 module in *Nicotiana*. We analyzed the preservation of the M4 module in *N. attenuata*, *N. miersii*, *N. pauciflora* and *N. obtusifolia*, of which we sequenced the transcriptomes of leaves induced by different HAEs to characterize transcriptional responses. The results revealed that the Z-summary scores of the M4 module, which indicate the level of module preservation, are all above 20 (values above 10 indicates that the module is highly conserved) for all pair-wise species comparisons, suggesting the M4 module has been retained among different species (*Table 1*). The statistical significance of the module preservation is further supported by permutation tests (in all comparisons, p<2.2E-16). We further analyzed the sequence divergence of M4 module genes by calculating the ω (Ka/Ks ratio) of M4 module genes shared between *N. attenuata* and *N. obtusifolia*, the two most divergent species in the dataset. The results revealed that the ω value of most M4 module genes (94.5%) are significantly less than 1 (p<0.05, Fisher's exact test, median ω=0.19), indicating they were under strong purifying selection. The distribution of ω from M4 module genes was not different from all leaf expressed genes (median ω=0.20, p=0.18, Wilcoxon–Mann–Whitney test), suggesting that the M4 module genes were not subject to strong divergent selection between *N. attenuata* and *N. obtusifolia*. Together, these results are consistent with the hypothesis that the identified M4 module is conserved among the different *Nicotiana* species.

**Table 1.** M4 module is highly preserved among four studied *Nicotiana* species. The number in each cell refers to the z-summary score calculated using 'modulePreservation' function from WGCNA package. Species in row and column indicate the reference and testing datasets, respectively. The score above 10 indicates the co-expression module is preserved, whereas the score bellow 2 indicate the module is not preserved. For all comparisons, p-values based on permutation tests are smaller than 2.2E-16.

|  | *N. obtusifolia* | *N. attenuata* | *N. miersii* | *N. pauciflora* |
|---|---|---|---|---|
| *N. obtusifolia* | - | 20.5 | 18.2 | 21.0 |
| *N. attenuata* | 29.4 | - | 34.9 | 31.2 |
| *N. miersii* | 22.8 | 38.8 | - | 27.8 |
| *N. pauciflora* | 23.4 | 26.2 | 27.4 | - |

The M4 co-expression module contains 1274 genes, which were enriched for gene ontology terms with 'regulation of defense responses', 'jasmonic acid metabolism', 'response to insects', and 'protein modification' among others (*Figure 5—figure supplement 1*). A majority of the JA biosynthetic genes were found in this module and their expressions were positively correlated with each other (*Figure 5*), indicating that the identified co-expressed genes reflect their functional relationships. Among all M4 module genes, 782 were significantly induced by FAC in the three species (*N. attenuata, N. acuminata* and *N. linearis*) which all showed JA accumulations 30 min after FAC elicitation. We infer that these 782 genes represent the core HAE-induced EDS network, which includes 75 protein kinases and 96 transcription factors (*Figure 5*). Previous research has shown that silencing genes in this conserved signaling network can directly affect herbivore-induced JA biosynthesis, metabolism and downstream defenses in *N. attenuata*. This includes the following protein kinase-encoding genes: *NaWIPK* (*Wu et al., 2007*), *NaMPK4* (*Hettenhausen et al., 2013*), *NaBAK1* (*Yang et al., 2011*) and *NaCDPK4*/5 (*Wu et al., 2007*; *Yang et al., 2012*), which are positive or negative regulators of JA biosynthesis and induced defense in *N. attenuata*; as well as the transcription factor, *NaWRKY6*, which is involved in differentiating mechanical wounding from herbivore attack and mediates plants' herbivore-specific defenses (*Skibbe and Galis, 2008*). Furthermore, several genes in this network have also been shown to be involved in phytohormone crosstalk and regulate JA-induced downstream defense responses, including *NaACO2, NaACS3a* and *NaETR1* which are involved in ET biosynthesis and perception (*von Dahl et al., 2007*); *NaLecRK* (*Gilardoni et al., 2011*; *von Dahl et al., 2007*) that inhibits SA accumulation during herbivory and *NaHER1* that suppresses abscisic acid (ABA) metabolism after herbivore attack, which, in turn, activates JA accumulation and defenses against insect herbivores (*Dinh et al., 2013*).

## A hub gene of the M4 module, *NaLRRK1*, forms a negative feedback loop with jasmonate signaling in the herbivore-induced EDS

Hub genes, which are defined as highly connected genes in the network, are often functionally important. Based on intra-modular connectivity, we identified 64 hub genes (top 5%) in the FAC-induced co-expression network, which include *NaWIPK*, a key positive regulator of JA biosynthesis and induced defense in *N. attenuata* (*Meldau et al., 2009*). To provide further mechanistic understanding of these hub genes in regulating induced defenses, we characterised an additional unknown hub gene encoding a putative leucine-rich repeat receptor kinase (*NaLRRK1*). The plasma membrane and nuclei localized *NaLRRK1* (*Figure 6a*) has an N-terminal extracellular region, a single transmembrane domain, and a C-terminal cytoplasmic region. The expression of *NaLRRK1* was co-upregulated with induced JA signaling among the six *Nicotiana* species (*Figure 6b*). Measuring *NaLRRK1* transcripts in leaves treated with different HAEs and one pathogen-associated elicitor, flg22, revealed that *NaLRRK1* is specifically induced by HAE (*Figure 6c*). We investigated whether HAE-induced JA signaling regulates the expression of *NaLRRK1* using two different jasmonate deficient transgenic plants, in which steps in JA signaling and perception were individually silenced (*Figure 6d*). Consistent with the microarray results, *NaLRRK1* was still significantly induced by FAC 30 min after elicitation in both of the JA-signaling deficient genotypes, revealing that JA is not

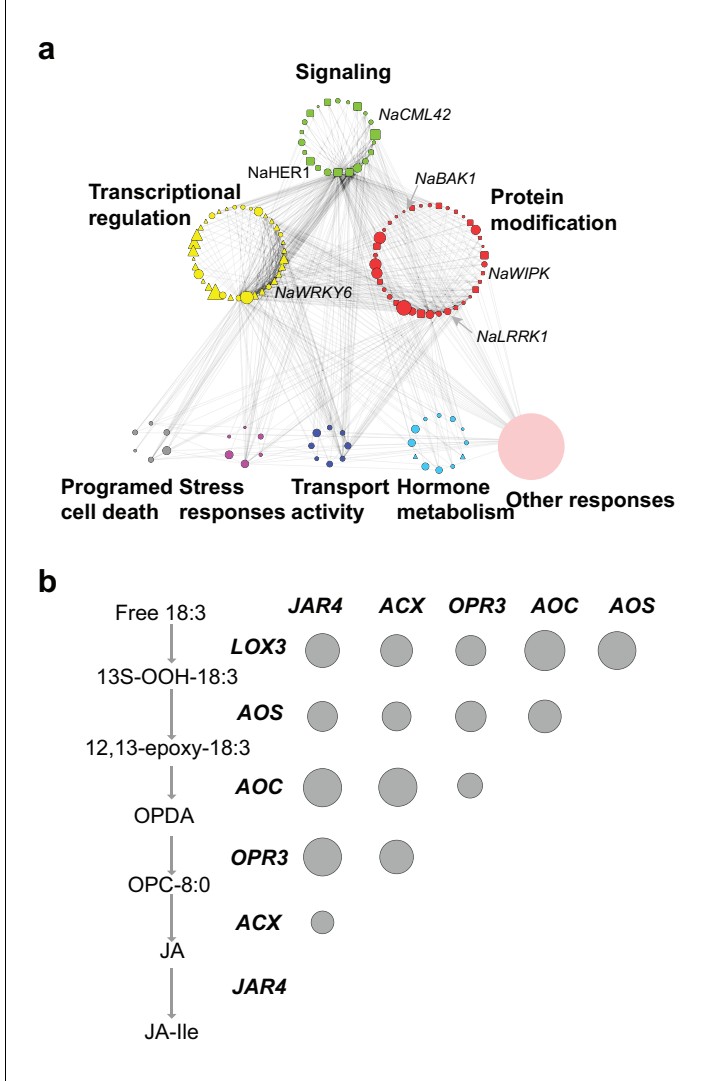

**Figure 5.** The co-expression network of module M4. (**a**) the network view of the M4 module. Each node represents a gene in the M4 module, except the filled orange node, which represents a collapsed node from a cluster. The shape of the node represents the property of the gene. Transcription factor: triangle, round rectangle: protein kinases, ellipse: other genes. The size of each node indicates their $log_2$ fold-change after FAC induction. The color of each node represents its Mapman functional annotation. Green: signaling, yellow: transcriptional regulation, red: post translational modification, gray: programmed cell death, purple: biotic and abiotic stress responses, dark blue: transport activity, light blue: hormone metabolism, orange: others. Edges represent the connections between two genes, estimated based on their co-expression coefficient. The genes that were shown to regulate HAE-induced anti-herbivore defenses are also shown in the network. (**b**) the correlation among genes involved in JA biosynthesis and metabolism. The left side shows biosynthesis and metabolism of JA, right side shows the correlation among each other. Each circle indicates the pairwise correlation coefficient between two genes. The size of the circle indicates the coefficient value. Only statistically significant correlations were shown ($p<0.05$, Pearson's product moment correlation test).

The following figure supplement is available for figure 5:

**Figure supplement 1.** Gene ontology (GO) enrichment analysis of the M4 module genes.

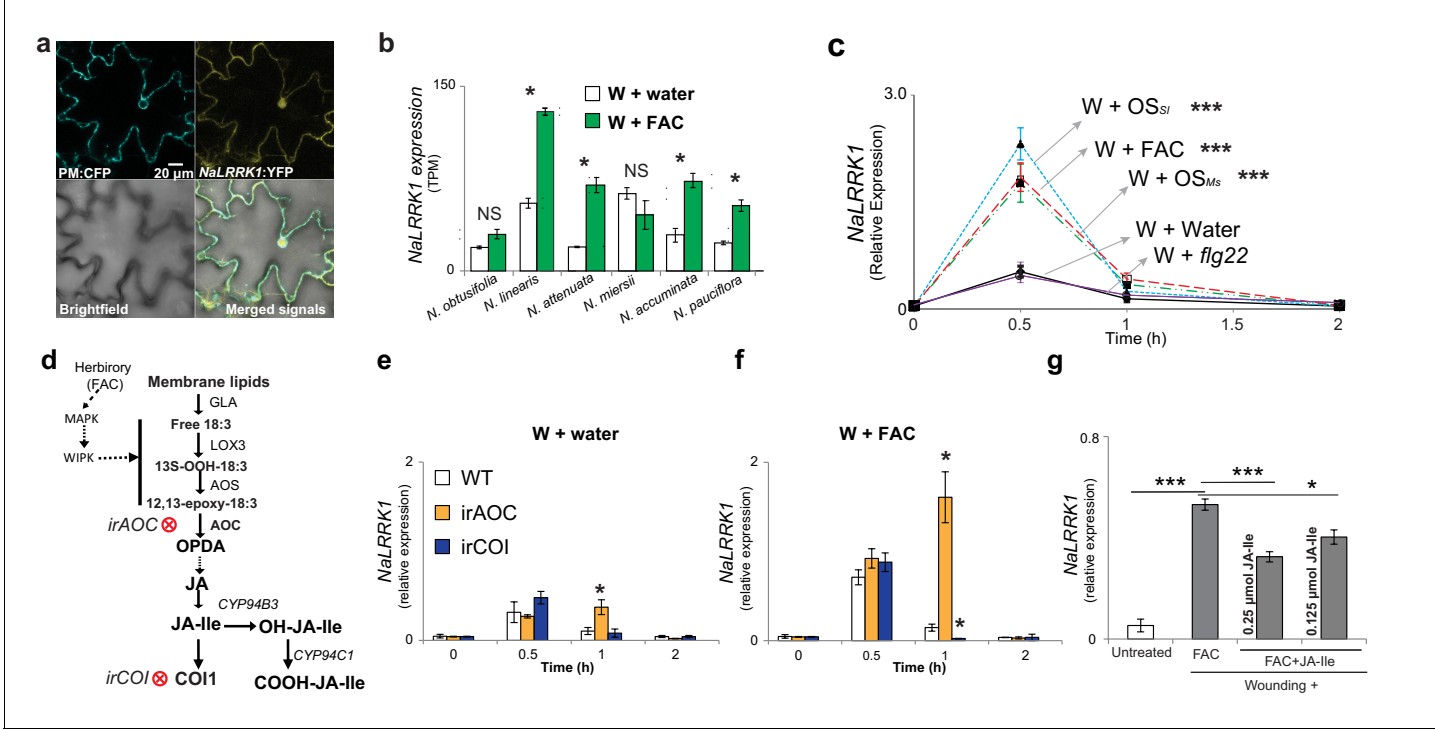

**Figure 6.** Jasmonate signaling suppresses the expression of *NaLRRK1*. (a) Subcellular localization of NaLRRK1. *Nicotiana attenuata* leaves were transformed with PM:CFP and NaLRRK1:YFP. After incubation for 48 hr, the transformed leaves were observed under a confocal microscope. The photographs were taken in UV light, visible light (bright field) and in combination (merged signals). Scale bar, 20 μm. (b) the transcript accumulation of *LRRK1* gene in the leaves of six *Nicotiana* species elicited by wounding + water (W + water) and wounding + FAC (W + FAC), estimated from RNA-seq data (n=3). Asterisk indicates FDR-adjusted p value <0.05 and fold change greater than 2. c, the kinetics of *NaLRRK1* transcript accumulation in *N. attenuata* leaves at 0, 0.5, 1 and 2 hr after treatments with different elicitors. For each treatment, 20 μL water or elicitors: FAC, oral secretions from *M. sexta* (OS$_{Ms}$), *S. littoralis* (OS$_{Sl}$), or flg22 were applied to the wounded leaves. Triple asterisks indicate the significant difference (p<0.01, n=5, except flg22 treatment was with 3 replicates) between treatment and control (W + water). d, a simplified model of JA biosynthesis and metabolism. The two transformed lines, in which AOC and COI were silenced respectively, are indicated. e and f, the transcript accumulation of *NaLRRK1* in the two transformed lines in comparison to WT after elicitation with water (e) or FAC (f). g, the FAC induced *NaLRRK1* transcript accumulation in *N. attenuata* leaves was suppressed by JA-Ile. *N. attenuata* leaves were collected at 1 hr after the induction. JA-Ile was applied in two different concentrations. For panel e, f and g, four biological replicates were used. In panel b, c, e f and g, data are presented as means ± SEM. Asterisk indicates significant difference (*: p<0.05; ***: p<0.01, Student's-t test) between treatments.

The following figure supplement is available for figure 6:

**Figure supplement 1.** In *N. attenuata*, OS$_{Ms}$ induced higher *NaLRRK1* transcript levels in 35S-*jmt*/ir-*mje* plants than in WT plants.

required for the up-regulation of *NaLRRK1*. Interestingly, compared to WT plants at 1 hr, *NaLRRK1* transcript levels were higher in *irAOC* plants – in which JA-Ile levels remain at basal levels - but were lower in *irCOI* plants, in which JA-Ile levels are constitutively high (*Paschold et al., 2008*) (*Figure 6e and f*). This indicates that JA-Ile levels may suppress the accumulation of *NaLRRK1* transcripts. To test this hypothesis, we compared *NaLRRK1* transcript accumulations in leaves in which the levels of JA-Ile were elevated by adding different amounts of JA-Ile to wounded leaves together with FAC. The results revealed that increased JA-Ile levels indeed decreased the levels of *NaLRRK1* transcripts (*Figure 6g*). Furthermore, in the transgenic plants 35S-*jmt*/ir-*mje*, in which endogenous JA levels are redirected to MeJA resulting in lower levels of induced JA-Ile and abrogated JA-signaling compared to WT plants (*Stitz et al., 2011*), HAE-induced *NaLRRK1* transcript accumulation was higher than in WT plants (*Figure 6—figure supplement 1*). These results are consistent with the hypothesis that JA-Ile negatively regulates *NaLRRK1* transcript levels.

We further investigated the roles of *NaLRRK1* in regulating HAE-induced defenses in *N. attenuata* using virus induced gene silencing (VIGS), which reduced HAE-induced *NaLRRK1* transcript

abundance by more than 88% in comparison to empty vector (EV) plants (*Figure 7—figure supplement 1*). The levels of a precursor of JA, OPDA, were significantly increased in VIGS-*NaLRRK1* plants

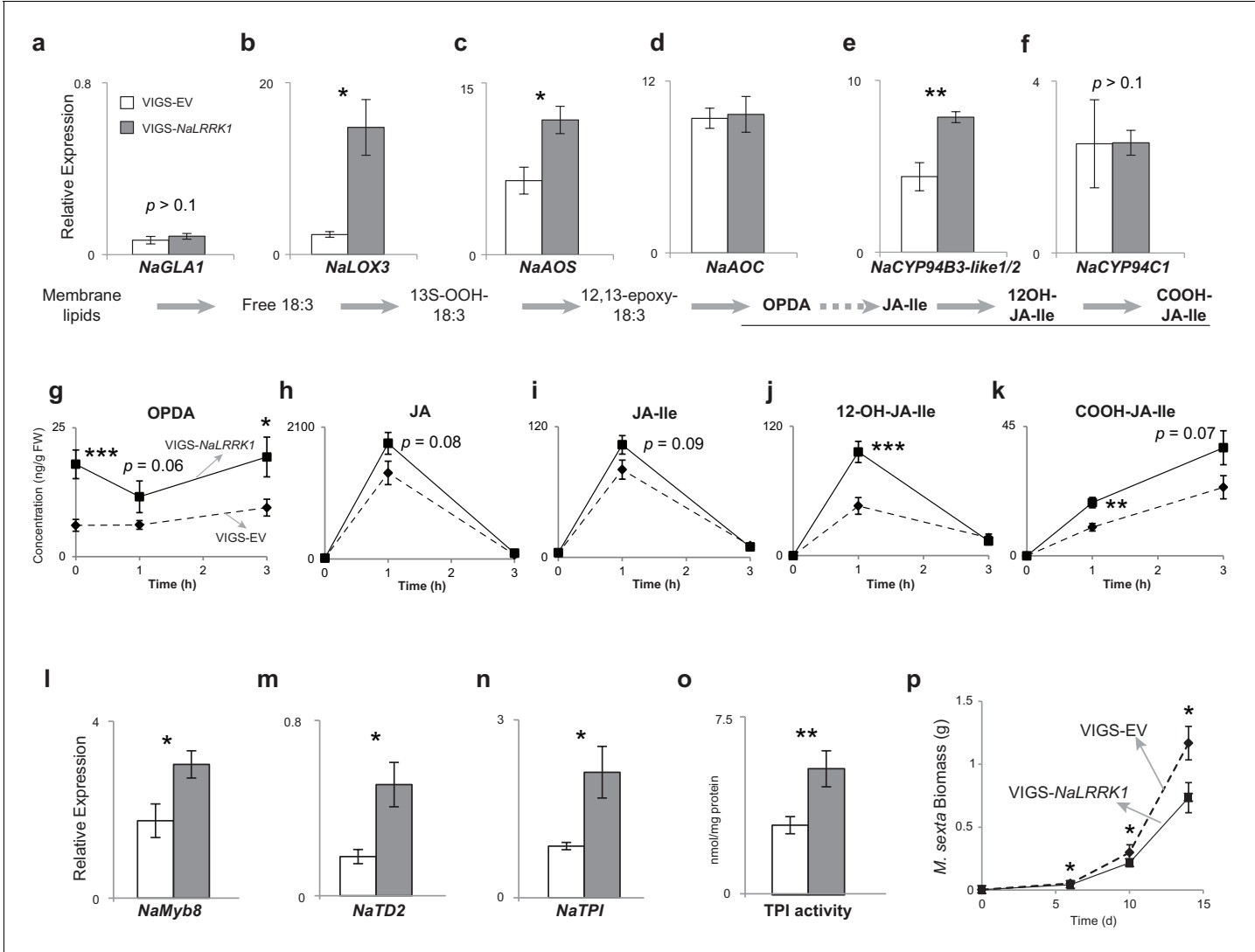

**Figure 7.** Silencing *NaLRRK1* increases FAC induced JA biosynthesis and metabolism and downstream defenses. (**a–f**) the VIGS-*NaLRRK1* plants have enhanced FAC-induced transcript accumulations of genes involved in JA biosynthesis and metabolism compared to EV plants (n=5). FAC elicitation significantly increased transcripts of: *NaLOX3* (**b**), *NaAOS* (**c**), *NaAOC* (**d**) and *NaCYP94B3-like1/2*. Transcripts levels were measured at 1h after FAC-elicitation. Due to high sequence similarity between *NaCYP94B3-like1 and NaCYP94B3-like2*, qPCR primers we used were not able to distinguish these two copies. **g-k**, the VIGS-*NaLRRK1* plants have enhanced JA biosynthesis and metabolism. FAC elicitation induces significantly higher levels of OPDA (**f**), OH-JA-Ile (**j**) and COOH-JA-Ile (**k**) in VIGS-EV plants than EV plants, but only marginally higher levels of JA (**h**) and JA-Ile (**i**) (n=7), **l**, the VIGS-*NaLRRK1* plants accumulated higher transcript levels of the transcription factor *NaMyb8* than did VIGS-EV plants. **m-o**, the VIGS-*NaLRRK1* plants accumulated higher transcript levels for the defense genes *NaTD2* (**m**) and *NaTPI* (**n**) and higher levels of TPI activity (**o**) than did VIGS-EV plants. (**p**) *M. sexta* gained significantly less mass when fed on VIGS-*NaLRRK1* plants than on VIGS-EV plants (n=24). The wounding + FAC treated leaf samples were collected at 1 hr after the treatment for gene expression analysis and at 24 hr after the treatment for TPI activity analysis. In all panels, data are presented as means ± SEM. Asterisk indicates significant difference (*, p<0.05; **, <0.01, ***, p<0.001, Student's-*t* test) between wounding + FAC treatment and control (wounding + water).

The following figure supplements are available for figure 7:

**Figure supplement 1.** *NaLRRK1* transcript abundance was successfully reduced in VIGS-*NaLRRK1* plants in comparison to controls.

**Figure supplement 2.** VIGS-*NaLRRK1* plants accumulated higher levels of FAC induced soluble sugars and invertase activity in comparison to control.

compared to EV (*Figure 7g*). Consistently, the transcript levels of genes involved in OPDA biosynthesis, such as *NaLOX3* and *NaAOS,* were all significantly increased in VIGS-*NaLRRK1* plants in comparison to EV plants (*Figure 7 a–d*), suggesting that *NaLRRK1* negatively regulates OPDA biosynthesis. Interestingly, the levels of JA and JA-Ile were not significantly different (*Figure 7 h and i*). However, both the levels of hydroxylated JA-Ile (12OH-JA-Ile) and transcripts of *NaCYP94B3-like1/2* - the homologue of *AtCYP94B3* that mediates hydroxylation of JA-Ile in *N. attenuata* (*Luo et al., 2016*) - were significantly increased in FAC-induced VIGS-*NaLRRK1* plants in comparison to VIGS-EV plants (*Figure 7e, f, j and k*). Since reduced expression of *NaCYP94B3-like1/2* results in lower levels of 12OH-JA-Ile and higher levels of JA-Ile (*Luo et al., 2016*), it is likely that the increased *NaCYP94B3-like1/2* transcript accumulations enhanced the hydroxylation of JA-Ile. These results suggest that *NaLRRK1* negatively regulates both JA biosynthesis and the hydroxylation of JA-Ile, and potentially suppress the effects the defense responses elicited by JA-signaling.

The examination of the downstream JA-dependent defensive traits in VIGS-*NaLRRK1* plants revealed that the net effect was a negative regulation of JA-signaling. In *N. attenuata*, the transcription factor *NaMYB8*, whose expression is activated by increased levels of endogenous jasmonates (*Kaur et al., 2010*), upregulates the expression of *NaTD2* and *NaTPI*, two key anti-herbivore defensive enzymes (*Kang et al., 2006b*). In comparison to VIGS-EV plants, VIGS-*NaLRRK1* plants accumulated significantly higher transcript levels of FAC-induced *NaMYB8, NaTD2* and *NaTPI* (*Figure 7 i-o*), consistent with the observed effects on JA signaling. In addition to *NaMYB8* regulated genes, FAC induced JA signaling also induces changes in primary metabolism, in particular soluble sugars and soluble invertases activity in *N. attenuata* (*Machado et al., 2015*). Here, we also found that VIGS-*NaLRRK1* plants showed higher levels of induced soluble sugars and invertases activity in comparison to those of VIGS-EV plants (*Figure 7—figure supplement 2*). Consistently, these higher levels of defensive responses in VIGS-*NaLRRK1* plants resulted in lower growth rates of *M. sexta* larvae in comparison to those feeding on VIGS-EV plants (*Figure 7 p*).

Taken together, the data suggest that the HAE-induced M4 gene *NaLRRK1* and jasmonate signaling form negative feedback loops that regulate and fine tune the induced defenses in *N. attenuata*.

## Whole genome duplications shaped the evolution of HAE-induced EDS network in *Nicotiana*

Having identified the M4 module, we were interested in exploring the evolution of HAE-induced EDS networks in *Nicotiana* by analyzing the evolutionary history of genes in the M4 module. For this, we first analyzed the most recent duplication event for each *N. attenuata* gene using the species reconciliation approach (Materials and Methods). Among all M4 module genes, 79.5% were retained in the genome of *Nicotiana* and *Solanum* after at least one round of duplication since the divergence of eudicots from monocots; this percentage retention is significantly higher than the genome-wide average (odd ratio = 1.96, p<2.2E-16, exact binomial test, *Table 2*). This suggests that gene duplications played a significant role in the evolution of the HAE-induced EDS network. Because Solanaceae taxa experienced a whole genome triplication (WGT) event (*Sato et al., 2012*), we compared the contributions of the Solanaceae WGT event with *Nicotiana*-specific lineage duplications (LD) to the evolution of the M4 module genes. A majority of the most recent duplication events of the M4 module genes occurred in the Solanaceae branch (51.5%), likely due to the WGT (*Sato et al., 2012*). This percentage is significantly higher than the genome-wide average (odd ratio = 1.46, p=1.79E-10, exact binomial test, *Table 2*) (*Figure 8*). Because our phylogenomic approach can not specifically distinguish the ancient segmental duplications from the WGT events, we further identified a subset of genes that is located in the syntenic blocks that resulted from the Solanaceae WGT. These genes are consistently significantly enriched in the EDS network (odd ratio = 1.40, p=2.4E-7). In contrast, only ~8.0% of the genes originated from *Nicotiana* lineage species duplications, which is not different from the genome-wide level (odd ratio=0.84, p=0.12, exact binomial test, *Table 2*). In addition, when considering genes that were significantly induced by FAC in all three species, *N. attenuata, N. acuminata* and *N. linearis* ('conserved EDS'), similar patterns were found (*Table 2*). These results suggest that Solanaceae WGT contributed more than lineage specific duplication events to the evolution of the HAE-induced EDS network.

Preferential gene retention followed by genome-wide duplications has been suggested as one of the major mechanisms for network evolution and expansion. Because the complete EDS network before the Solanaceae WGT is unknown, it is difficult to directly test the preferential gene retention

**Table 2.** Genes from multiple copy gene families and genes containing DTT-NIC1 TE insertions within 1kb upstream region are significantly enriched in the Nicotiana EDS network. The total number of genes used to test gene duplications and the DTT-NIC1 insertions analyses differed due to the additional filtering processes used in the former analysis. For the gene duplication analysis, we excluded all genes whose most recent duplication event was uncertain. WGT: whole genome triplication; NLD refers to *Nicotiana* lineage specific duplications; complete EDS refers to all of genes identified in the M4 module; conserved EDS refers to M4 genes that were significantly induced by FAC in all three species, *N. attenuata, N. acuminata* and *N. linearis*; genome-wide patterns were calculated based on all of genes that were expressed in *Nicotiana* leaves (normalized FPKM greater than 5 in at least three samples). Bold font color highlights the statistically significant values. Odd ratios were calculated by the following formula: Odd = (p1/[1 − p1])/(p2/[1 − p2]), where p1 is the percentage of genes that are part of the EDS network among testing group, e.g., genes from multiple gene families or genes retained from Solanaceae WGT, and p2 is the percentage of genes that are part of EDS network among all leaf expressed genes.

| | | # genes from multiple copy families | # genes retained from Solanaceae WGT | # genes retained NLD | Total number of genes after filtering |
|---|---|---|---|---|---|
| Genome wide | # genes | 9691 | 6181 | 1249 | 14,642 |
| Complete EDS | # genes | 906 | 587 | 87 | 1140 |
| | Odd ratio | **1.97** | **1.45** | 0.86 | |
| | p value | **< 2.2E-16** | **3.30E-10** | 0.17 | |
| Conserved EDS | # genes | 561 | 355 | 65 | 692 |
| | Odd ratio | **2.18** | **1.45** | 1.07 | |
| | p value | **< 2.2E-16** | **1.71E-06** | 0.41 | |

hypothesis. Instead, we examined a prediction that would result from the preferential gene retention scenario: for a given gene pair, the observed number of gene pairs that are both found in the EDS is higher than the number of two genes found in the EDS by chance. To test this prediction, we identified a subset of gene pairs (4292 pairs including 8584 genes) that resulted from the Solanaceae WGT (duplications on the Solanaceae branch) in which at least two of the three original copies were retained in the *N. attenuata* genome and expressed in leaves. Because both members of these gene pairs were both retained in the genome, we specifically examined the process of preferential gene recruitment to the M4 module. Among this subset of genes, 428 were found in module M4, and of those, both members of 120 gene pairs were in the M4 module. This is significantly higher than the expected number from the null model which assumes independent recruitment of two duplicated copies in the M4 module (p<2.2E-16, $\chi^2$ test), consistent with the prediction that duplicated genes were preferentially retained in the M4 module. Annotations of 120 gene pairs showed that more than 30.8% were either transcription factors or protein kinases, the proportion of which was significantly higher than by chance (2.56% among 4292 pairs, p<2.2E-16, binomial test).

Consistent with the genome-wide analysis, most of the M4 module genes that were previously functionally characterized in *N. attenuata*, including *NaHER1, NaCDPK*4/5, *NaMEK2, NaETR1* and *NaJAR4/6, NaACO1/2* evolved from the Solanaceae WGT event (*Figure 8*), and their homologs were also known to be involved in EDS in *Arabidopsis* (*Dong et al., 2004*; *Jung et al., 2007*; *Ludwig et al., 2004*; *Mewis et al., 2005*; *Romeis and Herde, 2014*; *Staswick and Tiryaki, 2004*) suggesting that their ancestral copies were likely already involved in EDS. Among these genes, the gene pairs of *NaHER1/2, NaCDPK4/5, NaJAR4/6* and *NaACO1/2* were highly co-expressed and all members were retained in the M4 module.

Taken together, these results showed that gene duplications, and likely preferential gene retention followed by WGT, shaped the evolution of *Nicotiana* HAE-induced EDS networks.

## Discussion

Regulation of herbivore induced defenses requires a complex and fine-tuned network (*Bonaventure et al., 2011*; *Erb et al., 2012*; *Wu and Baldwin, 2010*), as its fitness cost/benefit ratio

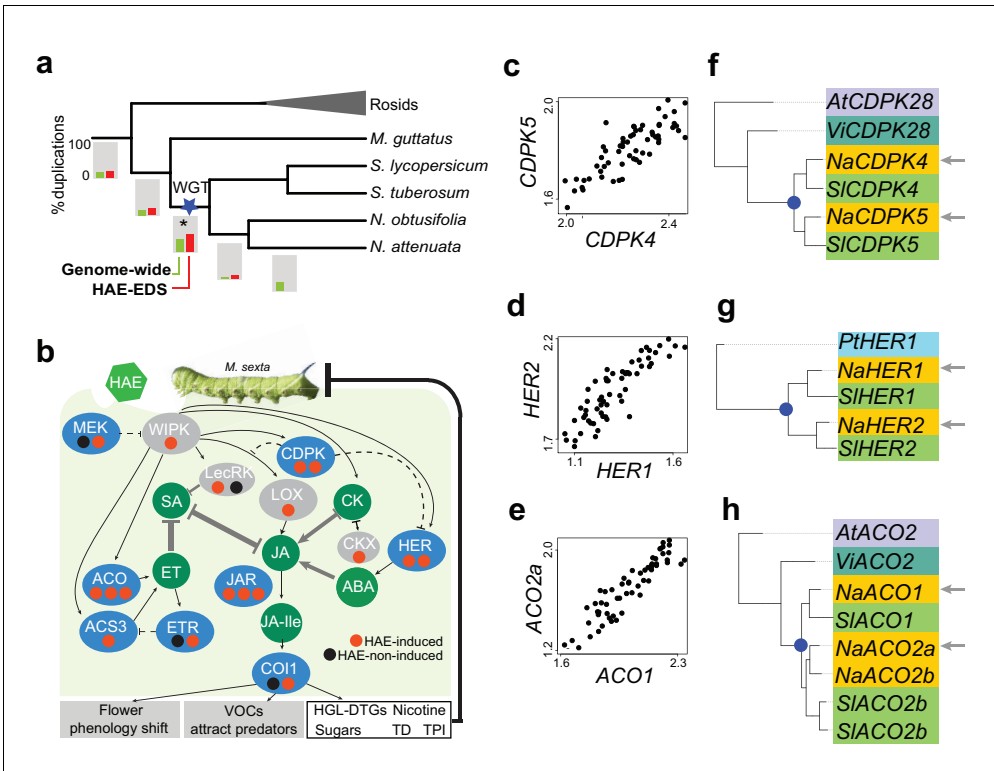

**Figure 8.** Solanaceae WGT contributed to the evolution of HAE-induced defense signaling in *Nicotiana*. (a) the gene duplication history of *Nicotiana attenuata* genes after the divergence of eudicots and monocots. Phylogenetic tree constructed based on one-to-one orthologue genes. The bars under each branch depict the percentage of duplications that occurred at a given branch. The green bar indicates the genome-wide (all genes expressed in leaves) pattern; red bar indicates the duplication events found in module M4. (b) WGT contributed to the evolution of genes that are involved in HAE-induced EDS. All genes are shown as ellipses, and phytohormones as circles. The color of every ellipse shows the most recent duplication events for each gene: blue and gray indicate the Solanaceae WGT, and ancient (shared with *Arabidopsis*) duplications, respectively. The dots underneath each gene represent the number of homologues found in the *N. attenuata* genome, and the color indicates whether the homologue was induced by FAC in *N. attenuata*. Red: significantly induced (FDR adjusted $p<0.05$, fold change greater than 1.5), black: not induced. Dashed lines indicate the indirect functional interactions. TD: *THREONINE DEAMINASE*; HGL-DTGs: 17-hydroxygeranyllinalool diterpene glycosides; TPI: trypsin proteinase inhibitor. (c–e) the co-expression patterns between the two homologous genes that likely resulted from the Solanaceae WGT; The expression values of each gene were from control and FAC-induced samples from the six *Nicotiana* species. (f–h) phylogenetic trees showing the duplication history of *NaCDPK4/5* (f), *NaHER1/2* (g) and *NaACO1/2* (h). The blue dot on the phylogenetic tree indicates the duplication events shared among Solanaceae species. The node colors indicate which species the homolog sequences belong to. *At* (lilac): *Arabidopsis thaliana*; *Vi* (turquoise): *Vitis vinifera*; *Pt* (light blue): *Populus trichocarpa*; *Na* (yellow): *N. attenuata*; *Sl* (green): *Solanum lycopersicum*.

depends on both the type of herbivore attacking the plant and the environmental context of the attack (*Baldwin and Hamilton, 2000*; *Glawe et al., 2003*; *Heil and Baldwin, 2002*; *Baldwin, 1998*). Therefore HAE-induced early defenses signaling that allows a plant to distinguish herbivore attack from wounding plays an important role in this process (*Bonaventure et al., 2011*; *Howe and Jander, 2008*; *Wu and Baldwin, 2010*). However, identifying herbivore induced EDS is challenging, due to its specificity among different tissues, time points, treatments and overall complexity (*Howe and Jander, 2008*; *Wu and Baldwin, 2010*). In this study, we took a novel comparative transcriptomic and co-expression network analysis approach using the leaf transcriptome data from six closely related species that were treated with different HAE, which resulted in the identification of a co-expression network that represents the HAE-induced EDS in *Nicotiana*. This approach assumes that

if two genes are functionally connected (co-expressed), the expression changes of one gene will also affect the other one during evolution, thus increasing the statistical power for detecting co-expressed gene modules. Although this assumption cannot be applied to species specific co-expression modules, it can be used to identify gene co-expression modules that are conserved among the studied species (*Table 1*), which likely are functionally important.

Using comparative transcriptomic and network analysis, we identified a co-expressed gene network (module M4), in which 782 genes represent the conserved HAE-induced EDS in *Nicotiana*. Large numbers of transcription factors and protein kinases were found in this network, suggesting rapid transcriptional and post-transcriptional regulations induced by HAE, which then likely led to the re-configuration of whole-plant metabolism to allow for the production of defense responses (*Gulati et al., 2013*). Interestingly, among these 782 genes in the HAE-induced EDS, of which only 28.2% and 11.7% in *N. obtusifolia* and *N. miersii* were elicited by FAC, respectively, at least 67.6% were induced by OS$_{SI}$ in each of these two species (*Figure 4*). These results suggest that the signaling network, while not elicited by FAC, remains intact in these two species, likely due to changes in FAC-perception. These results are also consistent with the analysis of module conservation which suggested that the M4 module is highly conserved among different *Nicotiana* species (*Table 1*).

Molecular signaling cascades often involve negative and positive feedback loops and form circuits. The M4 module is always co-activated and likely upstream or parallel to the activation of JA signaling, indicating that M4 module and JA signaling are likely involved in such circuits. We found *NaLRRK1*, a FAC-induced hub gene from the M4 module, and jasmonate signaling form negative feedback loops and are co-activated by HAE elicitation among different species that diverged several millions of years ago. The conserved co-activation between jasmonate signaling and *NaLRRK1* by HAE and its negative effect on insect performance suggests that the identified feedback loops are important for plant fitness in *Nicotiana*. Although increased expression of *NaLRRK1* after HAE elicitation may lower anti-herbivore defenses, it may increase the net fitness by reducing fitness costs associated with induced jasmonate signaling, such as the changes in primary metabolites that are important for a plant's tolerance of tissue removal and regrowth (*Machado et al., 2013b*). Clearly more components are involved in the *NaLRRK1* and jasmonate signaling negative feedback loops, such as transcription factors and other protein kinases, which are also likely present in the M4 modules. Future molecular studies that identify direct interacting components with *NaLRRK1* will shed light on the mechanisms of the *NaLRRK1*- JA feedback loops. The challenge will be to quantitatively analyze both transcriptional and post-transcriptional regulation of candidate genes at different time points after elicitation, since their interactions might be highly specific.

Analyzing the gene duplication history of the genes in the M4 module suggested that preferential gene retention after the WGT shared among *Nicotiana spp.* and *Solanum spp.* likely have contributed to the evolution of HAE-induced EDS in *Nicotiana*. We found that more than 30.8% of duplicated pairs that resulted from the Solanaceae WGT, of which both copies were retained in the M4 module, are either transcription factors or protein kinases. This proportion is significantly higher than by chance (p<2.2E-16), consistent with the dosage compensation hypothesis, which predicts that dosage-sensitive genes, of which transcription factors and protein kinases are examples, are more likely to be retained in the signaling network after genome-wide duplications (*Edger and Pires, 2009*; *Freeling, 2009*; *Hakes et al., 2007*; *Maere et al., 2005*; *Wapinski et al., 2007*).

Duplicated genes retained in the same network were often considered as evidence of functional redundancy (*De Smet and Van de Peer, 2012*; *Veron et al., 2007*); however, genetic redundancy is often evolutionarily unstable and is unlikely to be maintained over long timescales (*De Smet and Van de Peer, 2012*). The Solanaceae WGT event can be dated to 91–52 million years ago (*Sato et al., 2012*), yet many of the duplicated gene pairs remain co-expressed after HAE-elicitation. This suggests that retaining these duplicated copies in the same network has been beneficial to plants, likely as a result of increased network complexity and robustness (*De Smet and Van de Peer, 2012*). This is consistent with the results of studies on examining the function of *NaCDPK4/5* and *NaJAR4/6* by simultaneously silencing either member of both copies in *N. attenuata* (*Kang et al., 2006*; *Wang et al., 2007*; *Yang et al., 2012*). When *NaCDPK4* and *NaCDPK5* were individually silenced, HAE-induced JA accumulations were not affected; silencing both copies increased JA accumulation upwards of 3-fold, which resulted in significant negative fitness effects when plants were not attacked (*Yang et al., 2012*). Therefore, retaining both *NaCDPK4* and *NaCDPK5* in the network may increase the robustness against the negative fitness effects resulted

from null mutations in the gene itself or its regulatory systems (*Gu et al., 2003*). Interestingly, the duplicated gene pair, *NaJAR4/6*, which catalyzes the formation of JA-Ile, showed additive effects on HAE-induced JA-Ile accumulations, since silencing each individual copy both resulted in reduced JA-Ile levels (*Staswick et al., 2002*) and reductions in the activation of downstream defense responses. Thus, retaining both copies in the network resulted in higher level of HAE-induced JA-Ile and defense responses.

In addition to gene duplications, expansions of transposable elements (TEs) can also contribute to the evolution of induced signaling networks. For example, in rice, the *mPing*, a miniature inverted-repeat transposable elements (MITE) family, rapidly expanded in specific strains and its insertions into the 5' flanking region rendered adjacent genes inducible by abiotic stresses by introducing *cis*-regulatory elements and/or epigenetic markers (*Naito et al., 2009*; *Yang et al., 2005*; *Yasuda et al., 2013*). Similarly, we observed that insertions of the DTT-NIC1, a Solanaceae specific MITEs family that contains a stress inducible *cis*-regulatory element, the W-box, into 5' regulatory regions of genes are significantly enriched among genes of the HAE-induced EDS network in *Nicotiana* (p=0.00049, Appendix 1). We speculate that the genome-wide expansions of DTT-NIC1 may have facilitated gene recruitment into the *Nicotiana* EDS network by introducing *cis*-regulatory elements into the 5' flanking regions of *Nicotiana* genes. However, several other mechanisms could also result in the same observation; for example, biotic stresses may mobilize TEs, which are more likely to insert into genes with open chromatin under stressed conditions. Future studies that measure the contribution of DTT-NIC1 insertions in the inducibility of the identified EDS genes by manipulating the DTT-NIC1 insertions sites are needed to examine these hypotheses.

## Materials and methods

### Sample collection and RNA-sequencing

Plant material was collected as previously reported (*Xu et al., 2015*). In brief, the seeds of six *Nicotiana* species were germinated and grown in a York chamber under a 16/8 hr light/dark, 26°C and 65% relative humidity regime until the rosette stage. *Manduca sexta* and *Spodoptera littoralis* oral secretions (OS) were collected on ice from larvae reared on *N. attenuata* plants until the 3rd-5th instar as previously described (*Halitschke et al., 2001*). To simulate herbivore attack, one leaf of each plant was wounded with a pattern wheel and 20 μL of 1:5 diluted $OS_{Ms}$, $OS_{Sl}$ or FAC (138 ng μL-1 C18:3-Glu) or water (as control) was added to the puncture wounds. All leaves were collected 30 min after elicitation, their mid-veins rapidly excised, flash frozen in liquid nitrogen and stored at −80°C until analysis. For each species and treatment, three biological replicates were used based on common practice of RNA-seq experiments. The phytohormone data for all samples were analyzed and published in (*Xu et al., 2015*).

Total RNA was extracted from ~100 mg aliquots of homogenized leaves that were used for phytohormone analysis (*Xu et al., 2015*) using Trizol (Thermo Fisher Scientifc, Germany) according to the manufacturer's protocol. All RNA samples were subsequently treated with RNase-free DNase-I (Thermo Fisher Scientifc) to remove all genomic DNA contamination. The mRNA was enriched using the mRNA-seq sample preparation kit (Illumina), and ~200 bp insertion size libraries were constructed using the Illumina whole transcriptome analysis Kit following the manufacturer's standard protocol (Illumina, HiSeq system). All libraries were then sequenced on the Illumina HiSeq 2000 at the sequencing core facility at the Max Planck Institute for Molecular Genetics, Berlin. On average, more than 35 million 50-nt paired-end raw reads for each sample were obtained. All raw reads are deposited in the NCBI short reads archive (SRA) under the project number PRJNA301787.

### Gene co-expression network and differential expression analysis

All raw sequence reads were trimmed using AdapterRemoval (v1.1) (*Lindgreen et al., 2012*) with parameters '–collapse -trimns -trimqualities 2 -minlength 36' before being used for transcriptome assembly. We mapped all reads to the *N. attenuata* genome (release v2) using Tophat2 (v.2.0.6) (*Trapnell et al., 2009*). We used parameter '–segment-mismatches 2 -read-gap-length 2 -m 0 -N 2' for *N. attenuata* RNA-seq reads and allowed more mismatches for the other five species with parameters '–segment-mismatches 3 -read-gap-length 5 -m 1 -N 7'. The mapping statistics are shown in *Supplementary file 1C*. The reads count matrix was then extracted from the bam files using HTseq

with parameters '-a 1 -t exon' (*Anders et al., 2015*). For both mapping and reads counting, *N. attenuata* gene models were used. We further simulated the sequence divergence and estimated the expression levels to evaluate the effects of sequence divergence on reads mapping and gene expression estimation. The analysis revealed that sequence divergence did not affect the overall gene expression estimations using our mapping protocol.

We constructed the co-expression modules using the R package weighted gene co-expression network analysis (WGCNA) (*Langfelder and Horvath, 2008*) based on the trimmed mean of M-values (TMM) normalized FPKM (Fragments per kilobase of transcript per million mapped reads) values. Because the expression values among samples clustered by species, we applied a parametric normalization to reduce the effects that resulted from the background expression differences among species using the 'Combat' function from the sva package (*Johnson et al., 2007*). The normalized wounding and FAC-induced samples were first used to calculate the soft connectivity and the top 5000 connected genes were selected for module construction. The power selection, module significance and intra-module connectivity analysis were performed according to the WGCNA tutorial (http://goo.gl/twg20K). We only selected the genes in module M4 with membership greater than 0.75 for downstream analysis. FAC-induced hub genes were defined as the top 5% most connected genes in FAC-induced transcriptomes. In total, 67 genes were selected as hub genes. For visualization, the M4 module was exported to Cytoscape (3.1.0) with the edge weight greater than 0.15 as a cutoff.

Preservation of the M4 module at co-expression levels among four different species was analyzed using the 'modulePreservation' function from WGCNA. All pairs-wise comparisons were performed based on RNA-seq data from samples treated with WW, $OS_{Ms}$, FAC and $OS_{Sl}$. The M4 module genes that were not expressed in a given species were excluded. In total, 99.4% of genes were used for the module preservation tests.

The ratio of Ka/Ks (ω) was calculated using one-to-one orthologue genes between *N. attenuata* and *N. obtusifolia* using KaKs calculator (*Zhang et al., 2006*) with 'YN' method. Because the calculations of ω is unreliable for gene pairs with extremely low Ks values, all gene pairs with Ks value less than 0.02 were excluded. In total 1182 genes pairs (88.9%) were used for the analysis.

We identified differentially expressed genes using the edgeR (*Robinson et al., 2010*) package based on the raw count data. Genes with greater than 1.5-fold change and FDR-adjusted p-values less than 0.05 were considered as differentially expressed. For both the gene co-expression network and differential expression analyses, we only considered genes that had a FPKM greater than 5 in at least three samples. The venn diagram analyses for differentially expressed genes among species were performed using the R package venneuler (*Wilkinson, 2011*).

## Gene duplication detection

To identify gene duplication events, we first assigned homologous groups (HG) using a similarity-based method. To do so, we used all genes that were predicted from 11 eudicot genomes (Xu et al, submitted). In brief, all-vs-all BLASTP was used to compare the sequence similarity of all protein coding genes, and the results were filtered based on the following criteria: E-value less than 1E-20; match length greater than 60 amino acids; sequence coverage greater than 60% and identity greater than 50%. All BLASTP results that remained after filtering were clustered into HGs using the Markov cluster algorithm (mcl) (*Enright et al., 2002*).

For each of the identified HGs, we constructed a phylogenetic tree using an in-house developed pipeline. In brief, we aligned all coding sequences for each HG using MUSCLE (v.3.8.31) (*Edgar, 2004*) based on translated protein sequences with TranslatorX (v.1.1) (*Abascal et al., 2010*). For all aligned sequences, all non-informational sites (gaps in more than 20% of sequences) were removed using trimAL (v1.4) (*Capella-Gutierrez et al., 2009*). Then, for each HG, PhyML (v. 20140206) (*Guindon et al., 2009*) was used to construct the gene tree with the best nucleotide substitution model estimated based on jModeltest2 (v.2.1.5) (*Darriba et al., 2012*) with the following parameters: -f -i -g 4 -s 3 -AIC -a. The support for each branch was calculated using the approximate Bayes method (PhyML).

Duplication events within each HG were predicted based on the reconstructed gene trees using a tree reconciliation algorithm, which compares the structure of species and gene trees to infer duplication events (*Page and Charleston, 1997*). This approach allows one to predict the history of gene duplication events at each branch of the species tree. To reduce the false positives, we only

considered tree structures with approximated Bayes support greater than 0.9 at all three closest branches for assignment of gene duplication events.

## Microarray hybridization and data analysis

We measured the FAC-induced gene expression changes in WT and JA-deficient plants (ir*AOC*) using microarray analyses. The WT samples were the same as the samples used for the RNA-seq analysis. The germination and growth conditions, FAC elicitation, sample collection and RNA extraction for the analysis of the ir*AOC* plants were the same as those described for the analysis of the WT plants. cDNA preparation and hybridizations were performed as described in Kallenbach et al. (*Kallenbach et al., 2011*). Quantile normalization and $log_2$ transformation was performed for all raw microarray data using the R package 'Agi4x44PreProcess' (http://goo.gl/TJnA6Q). Probes with 1.5-fold change and adjusted *p*-values less than 0.05 were considered differentially expressed. The sequences of all probes were mapped to the *N. attenuata* draft genome (v1.0), and only the probes that uniquely mapped to annotated gene regions were considered for downstream analysis. All microarray data were deposited in the public GEO (Gene Expression Omnibus) repository (GSE75006).

## Functional annotation of genes

The gene functional annotation process was part of the *N. attenuata* genome sequencing effort (Xu et al, submitted). Multiple annotation tools were used. In brief, BLAST2GO (*Gotz et al., 2008*) was used to annotate the GO terms for all predicted genes, and the GO enrichment analysis was performed using the ClueGO (v2.1.1) (*Bindea et al., 2009*) plugin in Cytoscape. In addition, all *N. attenuata* genes were annotated using MapMan (*Thimm et al., 2004*) with annotation information from *Arabidopsis*, tomato, potato and cultivated tobacco. The transcription factors and protein kinase containing genes were identified based on the identified domains in each gene according to the criteria described in Pérez-Rodríguez et al. (*Riano-Pachon et al., 2007*) using the iTAK tool (http://bioinfo.bti.cornell.edu/cgi-bin/itak/index.cgi). All *N. attenuata* genes from NCBI were retrieved and compared to the predicted *N. attenuata* genes using BLAST and the best hits were annotated accordingly. The functional annotation of all *N. attenuata* genes and *N. attenuata* genome data are available from the *N. attenuata* database server (http://nadh.ice.mpg.de/NaDH/). The R scripts used for this study and original data are available as source code file and source data.

## qPCR confirmation of gene expression

Total RNA was extracted from ~50 mg leaves using Trizol (Thermo Fisher Scientifc) according to the manufacturer's protocol. In brief, all RNA samples were subsequently treated with RNase-free DNase-I (Fermentas) to remove all genomic DNA contamination. All cDNA samples were synthesized from ~1 µg total RNA using SuperScript II reverse transcriptase (Thermo Fisher Scientifc). The relative transcript accumulation levels of selected genes were measured using qPCR on a Stratagene MX3005P PCR cycler (Stratagene). For all qPCRs, the elongation factor-1A gene (*NaEF1a*, accession number: D63396) was used as the internal standard for normalization as previously described (*Johnson et al., 2007*). The primer pairs for qPCR are listed in *Supplementary file 1D*. All qPCR reactions were performed using qPCR core kit for SYBR Green I (Eurogentec) in a 20 µL reaction system. At least four biological replicates were used for all qPCR measurements.

## Characterizing the regulation of *NaLRRK1* expression

To characterize the expression of *NaLRRK1*, WT plants and three transformed lines were used: irAOC (line A-457) (*Kallenbach et al., 2012*) irCOI (line A-249) (*Paschold et al., 2007*) and 35S-*jmt*/ir-*mje* (line A-204) (*Stitz et al., 2011*). Seed germination procedure was the same as described above. Seedlings were transferred to Teku pots ten days after germination and then were planted into 1L pots in the glasshouse, which was maintained at 26–28°C under 16 hr of light as described in (*Krügel et al., 2002*).

The FAC elicitations were the same as described above. For the flg22 treatment (W+flg22), 20 µL of 100 nM flg22 in water was immediately applied to standardized puncture wounds produced by the fabric pattern wheel. To test the effects of JA-Ile on the expression of *NaLRRK1*, 0.25 $\mu$M or 0.125 $\mu$M JA-Ile in 20 $\mu$L FAC (containing 12.5% ethanol), was immediately applied to the puncture

wounds in leaves. The leaf samples (n=5) were collected 1 hr after treatment. All leaf samples were flash frozen in liquid nitrogen, and stored at −80°C until analyzed.

## Construction of reporter fusions and subcellular localization

The construction of NaLRRK1-YFP reporter fusion was carried out as described by Earley et al. and Ran et al. (*Earley et al., 2006*; *Li et al., 2014*), and a reporter fusion was also constructed for the *A. thaliana* plasma membrane (PM) intrinsic protein 2a (accession number: X75883) which was previously characterized as a marker for membrane associations (*Shibata et al., 2016*). The open reading frame (ORF) of *NaLRRK1* and PM were firstly amplified with Phusion Green High-Fidelity DNA polymerase (Thermo) by primer pairs listed in the *Supplementary file 1D*; an additional sequence (CACC) was then introduced into the forward primers to facilitate directional cloning into the pENTR/D-TOPO vector (Thermo). The reconstructed plasmids were transformed into *E. coli* TOP10 competent cells, then amplified and isolated as the 'entry vector' for the Gateway cloning. The 'entry vector' containing the ORF of *NaLRRK1* or *PM* was recombined into destination vectors using LR clonase (Invitrogen) to form a C-terminal NaLRRK1-YFP and C-terminal PM-CFP. Recombined plasmids were transformed into *E. coli* TOP10 competent cells, and then transformed into *A. tumefaciens* strain GV3101 for subsequent plant transformation. The transformation was performed using *A. tumefaciens* strain GV3101 following the protocol by Green et al. (*Green et al., 2012*). Fluorescence was visualized 48 hr following the inoculation with a Zeiss LSM 510 Meta confocal microscope (Carl Zeiss, Jena, Germany). The images were analyzed using LSM 2.5 image analysis software (Carl Zeiss, Inc.).

## Virus induced gene silencing (VIGS) of candidate genes expression and phytohormone measurements

VIGS based on the tobacco rattle virus (TRV) was used to transiently knock down the expression of the candidate gene *NaLRRK1* in *N. attenuata* as previously described (*Galis et al., 2013*). In brief, fragments of ~300 bp of target genes were amplified by PCR with primers listed in *Supplementary file 1D*. PCR fragments were recovered by agarose gel electrophoresis and purified using a gel band purification kit (Amersham Biosciences) according to the manufacturer's instructions, and subsequently digested with *Bam*HI and *Sal*I and inserted into plasmid, pTV00 (RNA1). After sequencing to validate the constructs, pTV-fragment-VIGS constructs and pTV00 (empty vector), together with RNA2, were transformed into *Agrobacterium* for the VIGS procedure.

At 21 days after *Agrobacterium* inoculation, rosette-stage plants were wounded with a pattern wheel and 20 µL of 1:5 diluted FAC (138 ng µL$^{-1}$ C18:3-Glu before dilution) or water was added to the puncture wounds. All samples were collected at 1 hr after elicitation with mid-veins excised, flash frozen in liquid nitrogen, and stored at −80°C until analysis. Silencing efficiency was quantified by qPCR. Overall, more than 88% of the target transcripts were silenced by VIGS.

Phytohormones were analyzed as described previously (*Wang et al., 2007*). In brief, ~100 mg frozen leaf was homogenized in a Genogrinder with 0.8 mL ethylacetate spiked with [9,10-$^2$H$_2$]-dihydro-JA and [$^{13}$C$_6$]-JA-Ile. Homogenates were centrifuged for 30 min at 4°C and the organic phase was collected and evaporated to dryness, which were subsequently reconstituted in 300 mL of 70% (v/v) methanol/water for analysis on an advance UPLC (Bruker), equipped with column ZORBAX eclipse XDB (Agilent) and quantified on an EVOQ triple quadrupole mass spectrometer (Bruker) using the MRM transitions described in (*Schäfer et al., 2016*).

## Secondary metabolites analysis and bioassay in VIGS plants

To quantify secondary metabolites that were known to function defensively in *N. attenuata*, leaves of VIGS-EV and VIGS-*NaLRRK1* (n=8) plants were treated with W+FAC for 24 hr, harvested and ground in liquid nitrogen and stored at −80°C until analysis.

Trypsin proteinase inhibitor (TPI) assay was carried out as previously described (*van Dam et al., 2001*). Briefly, 100 mg of ground powder (n=6) was extracted in a protein extraction buffer. The protein content was determined using the Bradford method and PI activity was analyzed with the radial diffusion assay, using soybean trypsin inhibitor (STI) as the external standard.

Soluble sugars (glucose, fructose and sucrose) and starch concentrations were quantified as described by Machado et al. (*Machado et al., 2013a*). Briefly, soluble sugars were extracted from

plant tissue (n=6) using 80% (v/v) ethanol, followed by an incubation step (20 min at 80°C). The precipitate was collected by centrifugation (15 min, 11,000 g, °C). Pellets were re-extracted twice with 50% (v/v) ethanol. Supernatants from all extraction steps were pooled and enzymatically quantified for sucrose, glucose and fructose. The remaining pellets were used for an enzymatic determination of starch.

To evaluate the performance of the specialist herbivore *M. sexta* on transformed plants, neonates were allowed to feed on EV and transformed plants (n=28) and their masses were measured at 0, 6, 10 and 14 d after transfer to experimental plants. To ensure that all larvae were at a similar developmental stage and had similar body mass at the start of the bioassay, newly hatched neonates were placed on untreated WT leaves for 48 hr and weighed. The neonates with similar size were selected for the bioassays.

35S-*jmt/ir-mje*: *N. attenuata* transgenic plants ectopically expressing *Arabidopsis (Arabidopsis thaliana)* jasmonic acid O-methyltransferase (35S-*jmt*) and with *N. attenuata* methyl jasmonate esterase silenced with RNAi.

## Acknowledgements

We thank T Krügel and the glasshouse team of the MPICOE for taking care of plants, P Langfelder for suggestions on data analysis, S Pottinger for help with sample collections, VIGS experiments and measure gene expression with qPCR, M Schäfer and M Kallenbach for setting up methods for the phytohormone analysis, R Li for helping with the cellular localization experiments and K Gase for helping with the RNA-seq data submission to NCBI. We thank J Wu, C Hettenhausen, M Huber and PM Schlüter for constructive comments on the manuscript and E Gaquerel for design suggestions of the *Figure 8b*. This work was supported by Swiss National Science Foundation (No: PEBZP3-142886 to SX), the Marie Curie Intra-European Fellowship (IEF) (No: 328935 to SX), Alfred and Anneliese Sutter-Stöttner foundation (to SX), the Max Planck Society and the European Research Council advanced grant ClockworkGreen (No. 293926 to ITB).

## Additional information

### Competing interests

ITB: Senior editor, *eLife*. The other authors declare that no competing interests exist.

### Funding

| Funder | Grant reference number | Author |
| --- | --- | --- |
| Schweizerischer Nationalfonds zur Förderung der Wissenschaftlichen Forschung | PEBZP3-142886 | Shuqing Xu |
| European Research Council | 293926 | Ian T Baldwin |
| European Commission | 328935 | Shuqing Xu |
| Max-Planck-Gesellschaft | | Wenwu Zhou<br>Thomas Brockmöller<br>Zhihao Ling<br>Ashton Omdahl<br>Ian T Baldwin<br>Shuqing Xu |
| Sutter-Stötner-Stiftung | | Shuqing Xu |

The funders had no role in study design, data collection and interpretation, or the decision to submit the work for publication.

### Author contributions

WZ, Acquisition of data, Analysis and interpretation of data; TB, Constructed phylogenetic trees and identified all gene duplications events for each gene family, Analysis and interpretation of data; ZL, Performed RNA-seq mapping and quantification and assisted the RNA-seq data analysis, Analysis and interpretation of data; AO, Performed motif analysis, Analysis and interpretation of data; ITB,

Drafting or revising the article, Contributed unpublished essential data or reagents; SX, Conception and design, Acquisition of data, Analysis and interpretation of data, Drafting or revising the article, Contributed unpublished essential data or reagents

**Author ORCIDs**
Ian T Baldwin, http://orcid.org/0000-0001-5371-2974
Shuqing Xu, http://orcid.org/0000-0001-7010-4604

## Additional files

### Supplementary files

• Supplementary file 1. Six supplementary tables. (A) Genes induced by FAC in all six *Nicotiana* species. Green and red colors indicate e3 ubiquitin-protein ligase and transcription factors, respectively. (B) Genes which showed lower expression in samples induced by *M. sexta* OS than by FAC in both *N. attenuata* and *N. pauciflora*. Numbers in each column indicate the TMM-normalized FPKM values. (C) Summary of RNA-seq reads from six closely related *Nicotiana* species. (D) Primers used for qPCR and VIGS in this study.

### Major datasets

The following datasets were generated:

| Author(s) | Year | Dataset title | Dataset URL | Database, license, and accessibility information |
|---|---|---|---|---|
| Shuqing Xu | 2016 | Evolution of the herbivore associated elicitor induced early defence signalling networks in Nicotiana | http://www.ncbi.nlm.nih.gov/geo/query/acc.cgi?acc=GSE75006 | Publicly available at the NCBI Gene Expression Omnibus (accession no: GSE75006). |
| Zhou W, Brockmöller T, Ling Z, Omdahl A, Baldwin IT, Xu S | 2016 | Evolution of the herbivore associated elicitor induced early defence signalling networks in Nicotiana | http://www.ncbi.nlm.nih.gov/bioproject/?term=PRJNA301787 | Publicly available at the NCBI BioProject (accession no: PRJNA301787). |

The following previously published dataset was used:

| Author(s) | Year | Dataset title | Dataset URL | Database, license, and accessibility information |
|---|---|---|---|---|
| Kim S, Gaquerel E, Gulati J, BaldwIN IT | 2011 | Tissue specific diurnal rhythm of transcripts and their regulation during herbivore attack in Nicotiana attenuata | http://www.ncbi.nlm.nih.gov/geo/query/acc.cgi?acc=GSE30287 | Publicly available at the NCBI Gene Expression Omnibus (accession no: GSE30287). |

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

**Appendix 1**

# Enrichment of the DTT-NIC1 in the regulatory region of the *Nicotiana* EDS genes

## Results

In addition to gene duplications, we further analyzed the putative role of TE insertions on the evolution of EDS in *Nicotiana*. Our previous work had revealed that several miniature inverted-repeat transposable elements (MITE) families, including DTT-NIC1, a Solanaceae-specific subgroup of the Tc1/Mariner family, expanded in *Nicotiana* (Xu et al, submitted). Permutation tests showed that insertions sites of annotated MITE families are enriched more than four times at 1kb upstream of genes compared to random chance (p<0.001, permutation test). To examine the role of MITE insertions on the evolution of the HAE-induced EDS, we compared the probability of genes recruited to the EDS that contain MITE insertions within 1 kb upstream to the genome-wide pattern. The results showed that MITE insertions into the 1 kb upstream regions significantly increased the probability of genes being recruited into the M4 module (odd ratio = 1.3, p= .0E-3, exact binomial test, *Appendix 1—table 1*).

**Appendix 1—table 1.** Insertions of the MITEs family DTT-NIC1 is significantly enriched in 1kb upstream region of M4 module genes. p-values were calculated based on exact binomial test. Superfamilies are represented using different letters in the names: DTT for Tc1/Mariner, DTM for Mutator, DTA for hAT, DTH for PIF/Harbinger. Statistically significant one is highlighted in red.

| MITEs name | Number of insertions to regulatory region of M4 module genes | Odd ratio | p-value |
|---|---|---|---|
| DTA_NIC1 | 4 | 0.54 | 0.32 |
| DTA_NIC2 | 3 | 1.14 | 0.75 |
| DTA_NIC3 | 22 | 1.38 | 0.16 |
| DTA_NIC4 | 3 | 0.55 | 0.49 |
| DTA_NIC5 | 1 | 0.23 | 0.19 |
| DTA_NIC6 | 4 | 0.68 | 0.66 |
| DTA_NIC7 | 1 | 0.72 | 1.00 |
| DTA_NIC8 | 1 | 0.61 | 1.00 |
| DTA_NIC9 | 12 | 1.42 | 0.28 |
| DTH_NIC1 | 23 | 1.53 | 0.063 |
| DTH_NIC2 | 23 | 1.14 | 0.56 |
| DTH_NIC3 | 11 | 1.28 | 0.38 |
| DTT_NIC1 | 65 | 1.63 | 0.0049 |

Analyses on the insertion of each individual MITE family separately revealed that the increased probability of EDS recruitment was mainly due to DTT-NIC1 to 1kb upstream region of genes (*Appendix 1—table 2* and *Supplementary file 1C*). In contrast, DTT-NIC1 insertions into downstream regions did not increase the likelihood of genes being recruited to the EDS (odd ratio = 1.1, p=0.5). Analyses of the genes that were significantly induced by FAC elicitation in all three FAC-responding species, *N. attenuata, N. acuminata* and *N. linearis*, showed similar patterns but had an even higher odd ratio (*Appendix 1—table 1*). These data suggest that insertions of DTT-NIC1 into gene regulatory regions significantly contributed to the recruitment of genes to the *Nicotiana* HAE-induced EDS.

**Appendix 1—table 2.** Seven DNA motifs are significantly enriched in the 1kb upstream region of M4 module genes. The presence of the motif in sequences and *p*-values were obtained using the HOMER package. Motif sequences were annotated by searching the plant *cis*-regulatory sequence database in PlantPan2 (http://plantpan2.itps.ncku.edu.tw/).

| Motif sequence | Annotation | Presence in M4 module genes (%) | Presence in background dataset (%) | p value |
|---|---|---|---|---|
| AGTCAACG | W-box1 | 60.5 | 38.5 | 1.0E-53 |
| TTGACCWT | W-box2 | 69.0 | 48.4 | 1.0E-47 |
| NNACGCGT | unknown | 19.7 | 10.0 | 1.0E-23 |
| TTCACACG | unknown | 18.7 | 11.6 | 1.0E-12 |
| GTCGGCTG | unknown | 39.5 | 29.7 | 1.0E-12 |
| CGAAGACT | unknown | 32.0 | 23.3 | 1.0E-11 |
| TTTCCACG | unknown | 34.3 | 25.9 | 1.0E-10 |

One of the mechanisms by which DTT-NIC1 insertions could contribute to the recruitment of a gene to the HAE-induced EDS is the introduction of promoter binding motifs. To examine this potential mechanism, we performed de novo identifications of enriched transcription factor binding motifs from regulatory regions among HAE-induced EDS genes. Within the 1kb upstream region sequences, seven significantly enriched motifs ($p < 1E-10$) were found among the M4 module genes in comparison to the genome-wide background (**Supplementary file 1D**). Among them, two W-box motifs (W-box1: AGTCAACG and W-box2: TTGACCWT) were the most statistically significant. These conserved motifs are known to be bound by WRKY transcription factors that are induced by biotic stresses (**Chen et al., 2012**; **Gao et al., 2016**; **Zou et al., 2011**). W-box1 and W-box2 box were found in the 1kb upstream regions of more than 60.5% and 69.0% of M4 module genes, respectively. A similar pattern was also found for the EDS genes that were induced by FAC in all three FAC-responding species.

Interestingly, at the genome-wide level, the distribution of the W-box1 motif in the *N. attenuata* genome is significantly associated with the insertions of DTT-NIC1 ($p < 0.001$, based on 1000 permutation test). Analyzing the consensus sequence of the DTT-NIC1 in *N. attenuata* also showed that DTT-NIC1 contains the W-box1 motif. Further comparisons of insertion sites of DTT-NIC1 and motif locations in the 1kb upstream region of *N. attenuata* HAE-induced EDS genes revealed that at least 44.4% of the DTT-NIC1 insertions introduced W-box1 to the regulatory region of adjacent genes. These results suggest that DTT-NIC1 mediated W-box insertions into the regulatory region of genes might have contributed to the recruitment of genes to the HAE-induced EDS network in *Nicotiana*.

## Method

We performed *de novo* motif identification using HOMER (**Heinz et al., 2010**), which identifies motifs that are significantly enriched in the test dataset in comparison to background datasets. The genes from the M4 module and leaf expressed genes that were not involved in the M4 module were used as positive and negative datasets, respectively. To reduce false positives, we performed additional analysis by identifying enriched motifs from 100 randomly selected subsets of genes from the negative dataset. These were then compared with the enriched motifs in the M4 module. We considered all of the enriched motifs from the M4

module that occurred in more than 5% of motifs identified from the random selected dataset as noise and removed them from downstream analysis. Using this approach, we identified seven 8 bp motifs (*Appendix 1—table 2*) that are significantly enriched in the regulatory region of M4 genes (p<1e-10). The presence and absence of the identified W-box motif in the *N. attenuata* genome were analyzed using the 'scanMotifGenomeWide' function from the HOMER package (*Heinz et al., 2010*).

Insertions sites of DTT-NIC1 to the regulatory region of genes were identified as described in (Xu et al, submitted). In brief, using all of the MITE consensus sequences as a library, we searched the 1kp upstream region of genes with RepeatMasker. The parameters '-nolow -no_is -s -cutoff 250' were used. The enrichment analysis of MITEs in 1kb upstream region of M4 module genes and associations between the MITEs and identified motifs were analyzed using permutation tests. For this, we randomly shuffled the insertion sites of MITEs 1000 times using the 'shuffle' function from the BEDTOOLS package (*Quinlan and Hall, 2010*). We then calculated the null models of MITEs insertion sites and overlaps between MITEs insertions and motif locations in the *N. attenuata* genome. These null models were then used to determine the enrichment significance of observed insertions of MITEs families in the 1 kb upstream region of M4 module genes as well as observed associations between locations of MITEs families and motifs.

