## [Decision Letter]

Thank you for submitting your article "Evolution of herbivore-induced early defense signaling was shaped by gene duplications and transposons in Nicotiana" for consideration by *eLife*.

Your article has been reviewed by two peer reviewers, and the evaluation has been overseen by a Reviewing Editor and Detlef Weigel as the Senior Editor. One of the reviewers, Dr. Anthony Bolger (Reviewer #1), has agreed to reveal his identity.

The reviewers have discussed the reviews with one another and the Reviewing Editor has drafted this decision to help you prepare a revised submission.

Summary:

Overall, this manuscript represents an impressive and comprehensive analysis of gene expression networks in response to herbivore induced stress. The authors use a comparative approach to identify well-conserved expression networks by examining expression patterns in multiple related Nicotiana species. The advantage of this approach is that it makes it possible to remove a great deal of noise. The authors demonstrate that one module, M4, is both highly conserved and is induced prior to JA induction. They also show that genes recruited to this module are much more likely to be retained following an ancient whole genome triplication in the Solanaceae, and are more likely to be associated with insertions of a particular transposable element in the region upstream of the gene. Overall, the expression analysis, along with associated hypothesis testing is impressive.

The reviewers were concerned with the analysis of duplicate genes and transposon insertions. While the first section of the paper represented a concise and carefully constructed set of experiments, the second section was far more speculative, somewhat internally contradictory and sometimes appeared to confuse correlation with causation. It read as if two very different papers had been combined into a single manuscript.

Essential revisions:

The reviewers found weaknesses with the interpretation of the TE analysis, especially on a gene-by-gene basis, although the correlation claim is relatively reliable on a genome-wide scale. Specifically, the causation claim seemed questionable. While targeted intervention experiments, such as removal/addition of a TE to could show the regulatory effect, such intervention experiments are unlikely to be feasible in a reasonable timescale. Another alternative would be natural variation between lines or closely related species, showing altered expression associated with the presence/absence of TEs, but this would require high quality genome sequences of suitable relatives, and these resources which may not exist.

The reviewers and editor recommend removal of the last Results section dealing with the TE analysis, and recommend that the authors paraphrase some of these results as speculation in the Discussion, along with the appropriate revision of title, Abstract etc.

---

## [Author Response]

[…]

*The reviewers were concerned with the analysis of duplicate genes and transposon insertions. While the first section of the paper represented a concise and carefully constructed set of experiments, the second section was far more speculative, somewhat internally contradictory and sometimes appeared to confuse correlation with causation. It read as if two very different papers had been combined into a single manuscript.*

*Essential revisions:*

*The reviewers found weaknesses with the interpretation of the TE analysis, especially on a gene-by-gene basis, although the correlation claim is relatively reliable on a genome-wide scale. Specifically, the causation claim seemed questionable. While targeted intervention experiments, such as removal/addition of a TE to could show the regulatory effect, such intervention experiments are unlikely to be feasible in a reasonable timescale. Another alternative would be natural variation between lines or closely related species, showing altered expression associated with the presence/absence of TEs, but this would require high quality genome sequences of suitable relatives, and these resources which may not exist.*

*The reviewers and editor recommend removal of the last Results section dealing with the TE analysis, and recommend that the authors paraphrase some of these results as speculation in the Discussion, along with the appropriate revision of title, Abstract etc.*

Thank you for the constructive comments and suggestions on our manuscript. We agree with the suggestions from the reviewers and editor regarding the TE analysis. In the revision, we removed the conclusions based on TE analysis from the Abstract and title and moved the results from the TE analysis to the appendix. To make this information available to readers, we briefly discuss the TE insertion patterns in the Discussion section, and speculate about the possible mechanisms.

Please note that during the revision, we updated our analysis with a newer version of *N. attenuata* genome annotation (v2.4), in which some redundant gene models were removed. The exact numbers changed slightly from the original version, but the results and conclusions remain unchanged from the earlier version.